# INTERVENTION-BASED CUMULATIVE CAUSAL FAIRNESS LEARNING

## ABSTRACT

Causal inference has emerged as a powerful framework for addressing algorithmic discrimination, offering a principled approach to understand and mitigate unfairness in decision-making systems. Various causality-based fairness notions have been proposed to quantify unfair causal effects stemming from sensitive attributes. Among these, intervention-based fairness has gained prominence as a foundational and widely applicable concept, computable from observational data. However, existing intervention-based fairness notions face critical limitations: (i) they fail to uniquely determine causal effects, and (ii) having zero value does not guarantee causal fairness. To overcome these drawbacks, we introduce a novel intervention-based fairness metric, the post-Intervention Cumulative Ratio Disparity (ICRD), to rigorously assess causal fairness. Building on this metric, we propose the Intervention-based Cumulative Causal Fairness Learning (`ICCFL`) framework, which trains causally fair decision models by generating interventional samples and computing differentiable approximations of ICRD. Theoretical analysis and empirical evaluations demonstrate that ICRD provides a robust measure for causal fairness. Extensive experiments on four benchmark datasets demonstrate that `ICCFL` significantly outperforms six state-of-the-art methods, improving fairness (MMD metric) over 40% on average, and effectively balancing fairness and accuracy.

## 1 INTRODUCTION

Fairness has become a crucial aspect in developing trustworthy machine learning algorithms. To address algorithmic discrimination, extensive efforts have thus been undertaken to quantify unfairness in those algorithms, giving rise the proposal of various fairness notions (Huan et al., 2020; Guldogan et al., 2023). Early fairness notions, such as demographic parity (Dwork et al., 2012; Jiang et al., 2020), and equalized odds (Hardt et al., 2016), are built on statistical correlations, and depend only on the joint distribution of decision model, sensitive attribute, covariates, and the outcome. However, these correlation-based approaches have a critical limitation, they cannot distinguish between discriminatory and spurious correlations between the outcome and sensitive attribute, leading to **inadequate assessments of fairness** (Kusner et al., 2017; Zuo et al., 2022).

To overcome these limitations, researchers have turned to causality-based fairness notions, where **causal fairness** is defined by the requirement that *the predicted outcome should be the same across the real-world without intervention and the counterfactual world with intervention on sensitive attribute* (Khademi et al., 2019; Galhotra et al., 2022). They encompass intervention-based notions (Huan et al., 2020; Ling et al., 2024) and counterfactual-based notions (Kusner et al., 2017; Pfohl et al., 2019). Counterfactual-based notions often require the full knowledge of causal model to compute nest counterfactuals (Didelez & Pigeot, 2001), which pose extra challenges compared to the ones based on interventions. As the most basic and general notion of causal fairness that is testable with observational data (Zuo et al., 2024), the intervention-based notions assess the unfairness as the causal effect of the sensitive attribute on the outcome. However, existing intervention-based fairness notions suffer from two key limitations: (i) they focus on mitigating the *average* causal effects in predictions, and thus **cannot consistently assess unfair effects over the entire predicted probability distribution**. (ii) **The value of these metrics being zero is not a sufficient condition** for a model to be causally fair. As shown in Table 2 and Figure S1 of our experiments, although

the value is low, there are noticeable differences in the predicted probability distributions across different sensitive groups.

To address these issues, we propose a novel causality-based fairness notion called post-Intervention-based Cumulative Ratio Disparity (ICRD), which measures the *cumulative causal effects* of interventions on sensitive attributes along predictive probabilities, providing a more robust assessment of causal fairness. We theoretically show that ICRD possesses several desirable properties, making it a more accurate and reliable metric. Building on this metric, we introduce the Intervention-based Cumulative Causal Fairness Learning (ICCFL) framework. ICCFL formulates a constrained optimization problem that integrates the ICRD metric into the model prediction loss. ICCFL leverages conditional multivariate normal distribution based on causal graph to generate interventional samples and employs a temperature-scaled Sigmoid function to approximate the intervention cumulative distribution function. By minimizing the cumulative distribution discrepancy induced by interventions on sensitive attributes and context, ICCFL effectively trains causally fair models. Our key **innovations** are given as below:

- We reveal that existing intervention-based metric cannot precisely assess violation of causal fairness, as they have fundamental limitations: i) their zero value does not guarantee zero discrimination; ii) their value does not accurately quantify the violation of model fairness.
- Based on above observations, we propose a novel fairness metric called ICRD to measure unfair effects along predictive probabilities, and derive several theoretical properties about the metric.
- We further develop a fairness framework called ICCFL. It leverages differentiable approximation of ICRD to train the fair decision model.
- Extensive experiments on four widely adopted benchmark datasets demonstrate that ICCFL significantly outperforms six state-of-the-art methods (Grgic-Hlaca et al., 2016; Kusner et al., 2017; Grari et al., 2023; Galhotra et al., 2022; Ling et al., 2024; Zuo et al., 2024), improving fairness over 40% on average with better (or comparable) accuracy. Our work advances fair machine learning by offering a practical and theoretically grounded solution for achieving causal fairness.

## 2 RELATED WORK

**Fairness Notions.** Causality-based fairness notions aim to assess the unfair causal effects of the sensitive attribute on decisions (Plecko & Bareinboim, 2023). Causal fairness notions can be categorized into intervention-based notions and counterfactual-based notions. The intervention-based notions measure unfair effects by performing interventions on the sensitive attribute. For example, Path-specific fairness (Zhang et al., 2018) assesses unfairness by measuring the causal effects transmitted along certain paths. $K$-Fair (Salimi et al., 2019; Ling et al., 2024) assesses the causal effects of sensitive attribute on the outcome under any assignment of admissible variables. Counterfactual-based notions examine how model predictions would change for individuals or subgroups under their counterfactual versions. For instance, counterfactual fairness (Kusner et al., 2017) requires that the predictions for the individuals should be the same as those for their corresponding counterfactuals, e.g., in recruitment decisions, it considers how a male candidate would be treated if he were female? However, some of these notions (Kusner et al., 2017; Zhang et al., 2018) require comprehensive knowledge of causal models or may encounter unidentifiable issues, where causal effects cannot be uniquely inferred from observational data. For example, counterfactual fairness is unidentifiable if there exists W-structure in the counterfactual graph (Wu et al., 2019). Moreover, *we point out that existing intervention-based notions are defined as heuristic criteria. They may underestimate unfairness in certain cases, as unfair effects across entire predicted probability distributions are diluted through averaging.*

To overcome the limitations of aforementioned notions, we introduce a novel metric ICRD to quantify the interventional fairness. We delineate its desirable properties and identifiable condition. Importantly, we prove that $ICRD = 0$ is a necessary and sufficient condition for causal fairness.

**Fair Machine Learning.** A number of fair machine learning methods have been proposed for various causality-based fairness notions (Schröder et al., 2024). These approaches can be broadly categorized into pre-processing, in-processing and post-processing mechanisms (Su et al., 2022). *Pre-processing* mechanisms (Jones et al., 2024) aim to detect and mitigate data bias presented before model training. *In-processing* mechanisms (Garg et al., 2019; Grari et al., 2023) enforce causality-based fairness constraints in the model training process to mitigate unfair causal effects.

*Post-processing* mechanisms (Wu et al., 2019; Mishler et al., 2021) rectify model predictions to mitigate unfair effects. Specifically, in the pursuit of interventional fairness, Galhotra et al. (2022) implemented conditional independence tests to identify the fair features that do not deteriorate intervention fairness. Ling et al. (2024) leveraged Markov blanket of the sensitive attribute to block the pathways between the sensitive attribute and variables to identify a subset of features that impact outcomes without violating intervention fairness. Zuo et al. (2024) incorporated the intervention fairness into a fairness-constrained optimization problem, allowing for a trade-off between accuracy and fairness. Despite the advancements, *existing methods cannot capture how sensitive attributes causally affect outcomes over the entire predicted probability distribution.*

In response, we propose `ICCFL` that incorporates a differentiable estimation of the proposed ICRD as a regularization term to promote fairness in classification. By tracing causal effects of sensitive attribute manifested along the model's predictive probability, `ICCFL` mitigates unfair impacts with principled optimization.

## 3 PRELIMINARIES

**Notation.** We use boldface uppercase $\mathbf{X}$ to describe a set of attributes, lowercase $\mathbf{x}$ to denote the values assigned to a subset of attributes, an attribute as $X$, and the value of an attribute as $x$. Let $\mathcal{D} = \{V_i = (\mathbf{S}_i, \mathbf{X}_i, Y) | 1 \le i \le n\}$ be a dataset with $n$ records, where $\mathbf{S}$ is the set of sensitive attributes, $Y$ is the decision attribute, and $\mathbf{X}$ denotes other observational variables. Specifically, we denote $\mathbf{C} \subseteq \mathbf{X}$ as the admissible context, through which the sensitive attribute is allowed to affect outcomes. We assume $\tilde{y} \in [0, 1]$ is the predicted probability of the decision model $h : \mathbb{R}^d \to [0, 1]$ with the model parameters $\theta$, and $\hat{y} \in \{0, 1\}$ is the corresponding binary prediction.

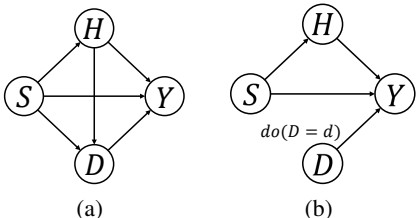

Figure 1: (a) is the ground truth causal graph; (b) is the causal graph with intervention on $D$.

**Causal Model.** Causal model can be formally expressed as a quadruple $\mathcal{M} = \langle \mathbf{V}, \mathbf{U}, P(\mathbf{U}), \mathbf{F} \rangle$, where $\mathbf{V}$ is the set of observable variables, $\mathbf{U}$ is the set of unobservable exogenous variables, $P(\mathbf{U})$ is the probability distribution over $\mathbf{U}$, and $\mathbf{F}$ is the set of causal structure functions $\mathbf{F} : \mathbf{U} \times \mathbf{V} \to \mathbf{V}$. A causal model is associated with a causal graph $\mathcal{G} = (\mathbf{V}, \mathbf{E})$, represented by a directed acyclic graph (DAG), which describes the causal interactions between variables. There is an edge from $V_i$ to $V_j$, i.e., $V_i \to V_j$, iff $V_i$ causes $V_j$.

**Intervention.** An intervention on $V_i \in \mathbf{V}$ (denoted by $do(V_i = v_i)$ or $\delta_{V_i=v_i}$) aims to break the causal function of variable $V_i$, and force $V_i$ to take a certain value $v_i$. Accordingly, all edges pointing to $V_i$ are discarded in the causal graph (e.g., Figure 1(b)). We denote $f(y|do(V_i = v_i))$ (or $f(y_{v_i})$ for short) as the post-intervention density of $Y$ affected by intervention $do(V_i = v_i)$. Specifically,

$$f(y|do(S = s)) = \int f(y|s, \mathbf{v}') \prod_{v \in \mathbf{v}'} f(v|pa(v; \mathcal{G})\delta_{S=s}) \mathrm{d}\mathbf{v}' \tag{1}$$

where $\mathbf{V}' = \mathbf{V} \setminus \{S, Y\}$, and $pa(v; \mathcal{G})\delta_{S=s}$ represents any term $pa(v; \mathcal{G})$ involved $S$, the value of $S$ is taken as $s$.

## 4 LIMITATIONS OF EXISTING INTERVENTION-BASED FAIRNESS NOTIONS

Interventional fairness requires that the model predictions should not be affected by interventions on sensitive attribute. To assess the violation of interventional fairness, several metrics has been developed, where path-specific fairness (Zhang et al., 2018) and $K$-Fair (Salimi et al., 2019) are widely-used.

Path-specific fairness assesses unfairness as causal effects of sensitive attribute on outcome transmitted along certain paths $\pi$:

$$PSF = |\mathbb{E}[\hat{y}_{s|\pi, s'|\bar{\pi}}] - \mathbb{E}[\hat{y}_s]| \tag{2}$$

$\hat{y}_{s|\pi,s'|\bar{\pi}}$ is the post-intervention outcome where the effect of intervention $do(s)$ is transmitted along $\pi$ while the effect of $do(s')$ is transmitted along other paths.

While $K$-Fair measures unfair effects of sensitive attribute on outcome on paths defined by admissible contexts $\mathbf{C}$:

$$KF = |\mathbb{E}[\hat{y}|do(S = s), do(\mathbf{C} = \mathbf{c})] - \mathbb{E}[\hat{y}|do(S = s), do(\mathbf{C} = \mathbf{c})]| \tag{3}$$

However, these fairness notions are insufficient for assessing the violation scores of causal fairness, since a lower value of them cannot accurately reflect the distribution disparity between sensitive groups. Here, we discuss the limitations of existing interventional fairness notions.

**Limitation 1: Inconsistent assessment.** Existing metrics assess the difference of intervention probability in average predictions. As a result, they are not capable of effectively assessing causal effects across the entire predictive probability distribution, which prevents them from accurately reflecting the differences in predictive probability distributions between different sensitive groups. In particular, in classification tasks, the values of these notions are highly associated with the selection of the decision threshold. To make a decision based on predictive probability, one predefined threshold is needed (Corbett-Davies et al., 2023). If the threshold for downstream tasks changes, the proportion of positive predictions of different groups will change accordingly, resulting in a change in the fairness score. The selection of the threshold greatly affects the value of existing intervention-based fairness notions. However, threshold tuning is needed in practice. Adjusting the threshold is a common practice for decision-making but can violate causal fairness if the model is evaluated using these metrics. For example, in college admissions, the number of admissions and applicants can vary from year to year, necessitating adjustments to the decision-making threshold.

Consider an example shown in Figure 1. In this example, the sensitive attribute is gender $S$, where $S = 0$ denotes female and $S = 1$ means male. The admissible context is department $D$, and the applicants' hobby $H$ is a proxy to $S$. $Y = 1$ indicates the applicant is admitted, while $Y = 0$ indicates rejection. Suppose the predicted probabilities of the model for Male group belonging to department 'A' is $\hat{y} = [0.4, 0.4, 0.6, 0.6, 0.5]$, and for Female group belonging to department 'A' is $\hat{y} = [0.3, 0.3, 0.5, 0.5, 0.9]$. If one sets $\alpha = 0.5$ for admission classification, we have $P(\hat{y} = 1|do(S = 1), do(D = \text{'A'})) = \sum_h P(\hat{y}|S = 1, D = \text{'A'}, h)P(h|S = 1) = P(\hat{y}|S = 1, D = \text{'A'}) = 0.6$, which is equal to $P(\hat{y} = 1|do(S = 0), do(D = \text{'A'}))$. Namely, existing interventional notions deem the model as fair. However, when $\alpha = 0.6$, $P(\hat{y} = 1|do(S = 1), do(D = \text{'A'})) - P(\hat{y} = 1|do(S = 0), do(D = \text{'A'})) = 0.4$, resulting in an opposite conclusion, which deems the model as unfair. In other words, **existing intervention-based notions cannot consistently measure the unfair effects of a model over the predictive probability distribution.**

**Limitation 2: Insufficiency.** Existing metrics are only the necessary but insufficient assessments for causal fairness. This is because their value being zero does not indicate that model's predicted probability distributions are the same across different sensitive groups. Their insufficiency stems from the reliance of existing intervention-based notions for causal effects on *average predictions* (first origin moment) to quantify discriminatory effects. However, according to probability theory (Bisgaard & Sasvári, 2000), two distributions being identical equal to having the same $r$-th origin moment for any $r$. We again consider the example discussed in **Limitation 1**. We can clearly observe that the model's predictions for admission probabilities reveal gender bias given admissible context $D$, because the model tends to give a higher admission rate for males and a lower admission rate for females. However, they cannot capture this discriminatory behavior, because $\mathbb{E}(\hat{y}|do(S = 1), do(D = \text{'A'})) = \mathbb{E}(\hat{y}|do(S = 0), do(D = \text{'A'})) = 0.5$. Figure S1(b) also shows that the predictive distribution gap between sensitive groups is evident despite the value of K-Fair is very close to zero. Thus, **zero-value of existing notions does not guarantee the model fairness.**

## 5 METHODOLOGY

### 5.1 THE PROPOSED FAIRNESS NOTION

To address the above limitations of existing intervention-based notions, we propose a new causality-based fairness notion called **Intervention-based Cumulative Rate Disparity** (ICRD for short). Specifically, ICRD aims to measure the cumulative causal effect of the sensitive groups on the model predictions. Its formal definition is as follows.

**Definition 1** (ICRD). *Given a set of admissible contexts* **C***, a decision model is considered as causality fairness if the following equation holds:*

$$\text{ICRD} = \int_0^1 |F(\tilde{y}_{s,\mathbf{c}})) - F(\tilde{y}_{s',\mathbf{c}})|\mathrm{d}\tilde{y} = 0 \tag{4}$$

*where $\tilde{y}$ is the model prediction probability, $F(\tilde{y}_{s,\mathbf{c}}) = P(\hat{y} \leq \tilde{y}|do(S = s), do(\mathbf{C} = \mathbf{c}))$ (where $\tilde{y} \in [0,1]$). It represents the cumulative distribution function of the model prediction intervened by the sensitive attribute $do(S = s)$ and context $do(\mathbf{C} = \mathbf{c})$.*

Note that the admissible contexts **C** denote those variables through which the sensitive attribute is permitted to influence the outcome. To make ICRD practical, the user is allowed to classify variables into admissible and inadmissible. Thus, admissible variables are part of the problem definition.

Compared to existing intervention-based fairness notions, our ICRD can more accurately capture the causal fairness of the decision model for its several advantageous properties, whose proof is given in Appendix.

**Theorem 1.** *The fairness notion ICRD has the following properties:* ❶ ICRD = 0 *iff the model predictions $\hat{y}$ are causally independent of the sensitive variable $S$ conditioned on any given context* **C** = **c***.* ❷ *The range of ICRD is within [0,1].* ❸ *ICRD is a continuous function.*

**Discussion.** ICRD provides a reliable and consistent assessment of the model fairness, and satisfies the sufficient condition for causal fairness, where $ICRD = 0$ indicates that the model is causal fairness. Let's reconsider the example discussed in **Limitation 2**. Unlike existing notions that fail to detect disparities in predictive distributions across sensitive groups and yield a value of zero, ICRD successfully uncovers such differences by measuring the cumulative causal effect of sensitive attribute on outcome across the entire prediction distribution, with $ICRD$=0.12. Therefore, ICRD can more accurately identify the discriminatory behavior of the model.

### 5.2 THE PROPOSED FAIR LEARNING METHOD

Based on the above analysis, we propose a novel fair learning method called `ICCFL`, which learns a decision model $h_\theta$ with the parameters $\theta$ to mitigate the cumulative causal effects of the sensitive attribute on predictions for achieving causal fairness. Specifically, `ICCFL` incorporates the ICRD metric as the fairness constraint into the prediction loss, which balances the inherent competition between accuracy and fairness. Formally, given a specific intervention on the admissible context $do(\mathbf{C} = \mathbf{c})$, the optimization function of `ICCFL` can be expressed as follows:

$$\min_\theta \frac{1}{n}\sum_{i=1}^n \ell(\tilde{y}^i, y^i) + \lambda|\text{ICRD}(\tilde{y})| \tag{5}$$

where the hyper-parameter $\lambda$ adjusts the trade-off between accuracy and fairness.

However, resolving the above optimization problem comes with the following challenges. (i) The assessment of cumulative post-intervention distribution $F(\tilde{y}|do(S = s), do(\mathbf{C} = \mathbf{c})$ in ICRD($\tilde{y}$) requires estimating stable conditional density function $f(x|pa(x, \mathcal{G}))$ and designing the model $\hat{y} = h(\mathbf{x}, \mathbf{s})$ to approximate $f(\hat{y}|\mathbf{x}, \mathbf{s})$. Unfortunately, computing ICRD of the distributions $F(\tilde{y}|do(S = s), do(\mathbf{C} = \mathbf{c}))$ and $F(\tilde{y}|do(S = s'), do(\mathbf{C} = \mathbf{c}))$ entailed by model $h$ is a tough task. (ii) It is quite impractical to perform direct regularization on ICRD, as ICRD is non-differentiable.

To solve challenge (i), `ICCFL` leverages conditional multivariate normal distribution to assess $f(x|pa(x, \mathcal{G}))$. Note that other conditional density estimation approaches can also be applied. As such, `ICCFL` can generate the interventional samples with each intervention $(do(S=s), do(\mathbf{C} = \mathbf{c}))$. Without loss of generality, we assume $\{\tilde{y}_s^{(1)}, \cdots, \tilde{y}_s^{(n_1)}\}$ with $n_1$ data points are the prediction probabilities for samples under intervention $(do(S=s), do(\mathbf{C} = \mathbf{c}))$, while $\{\tilde{y}_{s'}^{(1)}, \cdots, \tilde{y}_{s'}^{(n_2)}\}$ with $n_2$ data points are the prediction probabilities for samples under intervention $(do(S=s'), do(\mathbf{C} = \mathbf{c}))$.

Then, `ICCFL` can evaluate ICRD($\tilde{y}$) in Eq. (5) as:

$$\text{ICRD}(\tilde{y}) = |\frac{1}{n_1}\sum_{i=1}^{n_1}\mathbb{I}(\tilde{y}_s^{(i)} \leq \tilde{y}) - \frac{1}{n^-}\sum_{i=1}^{n_2}\mathbb{I}(\tilde{y}_{s'}^{(i)} \leq \tilde{y})| \tag{6}$$

where $\mathbb{I}(x)$ is the indicator function.

However, Eq. (6) is not differentiable with respect to the model parameters, causing difficulty in optimization. To solve the non-differentiable challenge (ii), `ICCFL` introduces a temperature-scaled Sigmoid function to perform a differentiable approximation mapping in Eq. (6) as follows:

$$\widehat{\text{ICRD}}(\tilde{y}) = |\frac{1}{n_1}\sum\nolimits_{i=1}^{n_1} \sigma_\tau(\tilde{y} - \tilde{y}_s^{(i)}) - \frac{1}{n_2}\sum\nolimits_{i=1}^{n_2} \sigma_\tau(\tilde{y} - \tilde{y}_{s'}^{(i)})| \tag{7}$$

where $\sigma_\tau(x) = \frac{1}{1+\exp(-\tau x)}$ is the mapping function, and $\tau$ is the temperature parameter. Notably, when $\tau$ tends to infinity, $\widehat{\text{ICRD}}(\tilde{y})$ converges to $\text{ICRD}(\tilde{y})$.

**Theorem 2.** *As $\tau \to \infty$, $\widehat{\text{ICRD}}(\tilde{y}) \to \text{ICRD}(\tilde{y})$.*

The proof is given in the Appendix. As a result, `ICCFL` can train a causally fair model by replacing $\text{ICRD}(\tilde{y})$ with $\widehat{\text{ICRD}}(\tilde{y})$ in Eq. (7). The overall procedure and complexity analysis of `ICCFL` are presented in Appendix.

### 5.3 IDENTIFIABILITY ANALYSIS

In practice, identifying the underlying causal graph of a real-world dataset can be quite challenging. Nonetheless, in most cases, we can derive a complete partially directed acyclic graph (CPDAG) using causal discovery methods. In this section, we discuss the minimum identifiability criterion for ICRD on an incomplete causal DAG. Specifically, we focus on the identification of the causal quantity $\int_0^1 F(\tilde{y}|do(\mathbf{S} = \mathbf{s}), do(\mathbf{C} = \mathbf{c}))d\tilde{y}$ in ICRD.

Similar to Zuo et al. (2024), we leverage the concept of partial causal ordering (PCO) (Perkovic, 2020) to provide the identifiability criterion for ICRD in the presence of CPDAG as follows.

**Proposition 1** (Identifiability Criterion for ICRD). *Let $\mathcal{G} = (\mathbf{V}, \mathbf{E})$ be a MPDAG where $\mathbf{V} = \{\mathbf{S}, \mathbf{X}\}$, and $\mathcal{G}^* = (\mathbf{V} \cup \{\hat{Y}\}, \mathbf{E})$ be the augmented $\mathcal{G}$, $\hat{Y}$ is the model output. Suppose $\mathbf{C} \subset \mathbf{X}$ is the admissible context and $PCO(\mathbf{V}, \mathcal{G}) = (\mathbf{B}_1, \cdots, \mathbf{B}_t)$ where $(\mathbf{B}_1, \cdots, \mathbf{B}_t) \subseteq \mathbf{V}$ are mutually disjoint sets. $\mathbf{B}_i < \mathbf{B}_j$ ($i < j$) satisfies the condition that there is at least an edge directed from $B_i \in \mathbf{B}_i$ to $B_j \in \mathbf{B}_j$ in $\mathcal{G}$. Iff there is no any possibly causal relationship between $O \in \mathbf{S} \cup \mathbf{C}$ and $V \in \mathbf{V}\backslash\{\mathbf{S}\cup\mathbf{C}\}$ such that $O - V$ is in $\mathcal{G}^*$, then for any density function $f(\mathbf{x})$ consistent with $\mathcal{G}$ and the conditional density $f(\hat{y}|\mathbf{x})$ consistent with $\mathcal{G}^*$, $ICRD$ is identifiable with $\mathbf{B}_i$.*

$$\int_0^1 F(\tilde{y}|do(\mathbf{s}),do(\mathbf{c}))d\tilde{y} = \int_0^1 \int_0^{\tilde{y}} \int f(\hat{y}|\mathbf{v}',\mathbf{s},\mathbf{c}) \prod_{\mathbf{b}_i \subseteq \mathbf{v}'} f(\mathbf{b}_i|pa(\mathbf{b}_i, \mathcal{G})\delta_{\mathbf{S}=\mathbf{s},\mathbf{C}=\mathbf{c}})d\mathbf{v}'d\hat{y}d\tilde{y}$$

$$\tag{8}$$

*where $\mathbf{V}' = \mathbf{V}\backslash\{\mathbf{S}, \mathbf{C}\}$, and values $pa(\mathbf{b}_i, \mathcal{G})$ means if the parents of $\mathbf{b}_i$ contain any $S \in \mathbf{S}$ or $C \in \mathbf{C}$, their values are specified by s or c.*

The proof is deferred to Appendix.

**Discussion.** The above proposition shows that our proposed metric **ICRD and** `ICCFL` **do not rely on the assumption of complete knowledge of the causal DAG.** When the graphical criteria outlined in Proposition 1 are satisfied, then ICRD is identifiable. This significantly expands the practical applicability of our metric and method, as partially DAGs can be directly obtained using existing causal discovery algorithms. For the scenarios where the causal effect of $\mathbf{S} \cup \mathbf{C}$ on $\hat{Y}$ is unidentifiable, we can compile all CPDAGs that align with each valid permutation of edge orientations between $O - V$, where $O \in \mathbf{S} \cup \mathbf{C}$ and $V \in \mathbf{V}\backslash\{\mathbf{S} \cup \mathbf{C}\}$. As a result, we can uniquely identify the causal quantity $\int_0^1 F(\tilde{y}|do(\mathbf{s}), do(\mathbf{c}))$ in each CPDAG, and replace the fairness constraint in Eq. (5) with the average of unfairness over these CPDAGs. We also remark that our `ICCFL` can easily extend to multiple sensitive attributes, as discussed in Appendix B.2.

## 6 EXPERIMENTAL RESULTS AND ANALYSIS

### 6.1 EXPERIMENT SETTINGS

**Datasets.** We conduct experiments to evaluate the effectiveness of our `ICCFL` using four real-world datasets (Law School, Adult, Dutch and ACSPublicCoverage). The Law school dataset consists

Table 1: Accuracy and fairness results (MMD) of each method on real-world datasets. The best results are highlighted with **bold**. '-' denotes the comparison is not applicable, due to the absence of any non-descendants of sensitive attribute. ○/● indicates that ICCFL is statistically worse/better than the compared method by student pairwise $t$-test at 95% confident level.

| | Adult | | Dutch | | Law School | | ACSPublicCoverage | |
|---|---|---|---|---|---|---|---|---|
| | Acc.↑ | MMD↓ | Acc.↑ | MMD↓ | MAE↓ | MMD↓ | Acc.↑ | MMD↓ |
| Baseline | 0.766○ | 48.974● | 0.784○ | 36.996● | 0.734○ | 78.836● | 0.788○ | 42.449● |
| Unaware | 0.765○ | 22.687● | 0.776○ | 20.075● | 0.746 | 36.244● | 0.779○ | 21.758● |
| A3 | 0.736● | 19.697● | 0.757 | 15.516● | 0.758 | 8.836● | 0.755● | 18.822● |
| ALCF | 0.751 | 14.702● | 0.772 | 11.207● | 0.748 | 6.324● | 0.762 | 10.594● |
| SeqSel | 0.726● | 12.953● | 0.748● | 10.255● | - | - | 0.744● | 8.907● |
| FairCFS | 0.725● | 12.788● | 0.746● | 8.446● | - | - | 0.736● | 8.142● |
| $\epsilon$-IFair | 0.744 | 13.583● | 0.765 | 10.496● | 0.752 | 6.104● | 0.768 | 9.698● |
| ICCFL | 0.746 | **5.745** | 0.763 | **3.916** | 0.750 | **3.174** | 0.766 | **4.189** |

of 20,412 records, where we treat '*race*' as the sensitive attribute, and 'entrance exam socres' as the context variable. We consider the causal graph introduced by Kusner et al. (2017) (level-2 causal model) as the ground truth. The ACSPublicCoverage is a large-scale dataset, with a total of 1,138,289 records. We treat '*race*' as the sensitive attribute and 'CIT' (nationality) as the context variable. We consider the causal graph introduced by Helwegen et al. (2020) (level-3 causal model) as the ground truth for the ACSPublicCoverage dataset. The Adult dataset consists of 48,842 samples with 11 variables, where we treat '*sex*' as the sensitive attribute, and 'education' as the context variable. The Dutch dataset contains 60,421 samples with 12 variables, where we also treat '*sex*' as the sensitive attribute, and 'country_birth' as the context variable. We follow (Zhang et al., 2018; Wu et al., 2019) to construct the causal graphs. Specifically, we employ the original PC algorithm (Spirtes et al., 2000) and set the significance threshold 0.01 for conditional independence testing in graph construction. The causal graphs of these datasets are given in the Appendix.

**Comparison.** The experiments are conducted by comparing ICCFL against: (i) **Baseline** uses all variables to train the model without considering the fairness; (ii) correlation-based method: **Unaware** (Grgic-Hlaca et al., 2016) uses all variables except the sensitive attribute to train the model; (iii) causality-based methods: **A3** (Kusner et al., 2017) assumes the causal model as the additive noise model, and assesses the noise term, which is then used to train the causal fairness model; **ALCF** (Grari et al., 2023) employs adversarial learning with a causal model to generate counterfactuals, and then trains the interventional fairness model based on the augmented data; (iv) interventional fairness-focused methods, **SeqSel** (Galhotra et al., 2022) applies conditional indenpendence tests to select the features that satisfy the interventional fairness, **FairCFS** (Ling et al., 2024) uses the Markov Blanket (MB) discovery method to identify the MB of sensitive attribute and outcome, and then selects fair features that are independent of sensitive attribute but relevant to outcome; $\epsilon$-**IFair** (Zuo et al., 2024) incorporates the interventional fairness into the training process. Each compared method uses the same two hidden layers ReLU neural network with 64 hidden neurons as the base model, so they have the same number of model parameters. Note that sensitive attributes are only required for training (i.e., fairness constraints and intervention simulations) in ICCFL.

**Evaluation metrics.** We use *Accuracy* for classification tasks and *mean absolute error* (MAE) for regression tasks to measure the prediction performance. Recall that a decision model is causally fair if there is no disparity in the distribution of prediction probabilities on different interventional samples generated by the ground truth causal model. As such, to evaluate different methods with respect to causal fairness, we use a *distribution-level* distance metric, i.e., Maximum Mean Discrepancy (MMD) (Shalit et al., 2017) to measure the distribution divergence.

**Hyper-parameters settings.** For all used datasets, we split the dataset into training, validation, and test sets with proportions of 70%, 10%, and 20%, respectively. We report the average results of ten random splits. As to the hyper-parameters for each compared method, we use the grid search strategy (ranges specified in Table A1 of Appendix) on the validation set to choose the best values. The hyper-parameter analysis of our ICCFL is also given in Appendix. Similar to Ma et al. (2023), we leverage widely-used Pyro (Bingham et al., 2019) to construct the causal models of the datasets.

## 6.2 PERFORMANCE COMPARISON

Table 1 presents the performance of compared methods in terms of accuracy and fairness. We can observe that: i) ICCFL outperforms compared methods in terms of fairness, and achieves a higher (or similar) accuracy than them. This indicates our ICCFL can effectively mitigate the unfair causal effects. ii) Compared to Baseline, both ALCF and $\epsilon$-IFair exhibit a reduction in fairness violation, and achieve an acceptable balance between fairness and accuracy. This suggests that utilizing traditional interventional fairness helps to mitigate the discrimination of model. However, compared to ICCFL, their lower performance in terms of MMD highlights the limitations of these approaches in achieving causal fairness.

Table 2: Accuracy and fairness results of ICCFL and its variants on real-world datasets. The better results of fairness are highlighted with **bold**. ○/● indicates that ICCFL is statistically worse/better than the compared variant by student pairwise $t$-test at 95% confident level.

| | Adult | | | |
|---|---|---|---|---|
| | Acc.↑ | $K$-Fair↓ | ICRD↓ | MMD↓ |
| Baseline | 0.766○ | 0.204● | 0.326● | 48.974● |
| ICCFL-wF | 0.769 | **0.174●** | 0.302● | 44.473● |
| ICCFL-KF | 0.744 | **0.036○** | 0.152● | 12.746● |
| ICCFL | 0.743 | 0.042 | **0.057** | **5.745** |
| | Dutch | | | |
| | Acc.↑ | $K$-Fair↓ | ICRD↓ | MMD↓ |
| Baseline | 0.784○ | 0.198● | 0.232● | 36.996● |
| ICCFL-wF | 0.786 | **0.193●** | 0.228● | 35.399● |
| ICCFL-KF | 0.766 | 0.027 | 0.138● | 10.268● |
| ICCFL | 0.763 | **0.016** | **0.022** | **3.916** |

In addition, among fairness-aware methods, SeqSel and FairCFS typically achieve lower predictive performance. This is because they exclude certain important features with information about the outcome to ensure fairness. A3 exhibits the worst trade-off between accuracy and fairness due to the unrealistic causal model assumptions. iii) Although Baseline achieves the highest accuracy, it performs the poorest in fairness. This is because the primary objective of Baseline is to optimize accuracy. In addition, Unaware mitigates discrimination by excluding the sensitive attribute, but still struggles to reduce unfair effects caused by descendants of the sensitive attribute. In contrast, ICCFL can mitigate the negative impacts of the sensitive attribute and its descendants by minimizing the cumulative causal disparity.

## 6.3 THE SUPERIORITY OF ICRD

To study the advantage of our ICRD metric for assessing the causal fairness, we consider two variants of ICCFL: i) ICCFL-KF incorporates $K$-Fair during the model training; ii) ICCFL-wF generates interventional samples but trains the model without fairness constraints. The results are presented in Table 2 and Figure S1 (which shows the cumulative distribution functions of model predictions for sensitive groups in Appendix). We have the following conclusions: i) ICCFL obtains clearly better ICRD re-

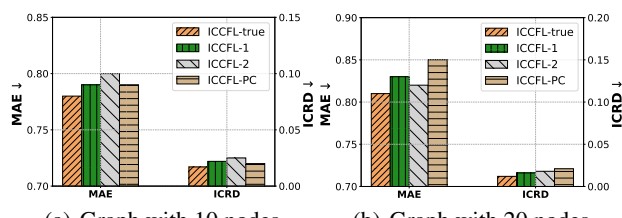

(a) Graph with 10 nodes     (b) Graph with 20 nodes

Figure 2: Accuracy and fairness trade-offs of ICCFL under different complete partially DAGs.

sults across real-world datasets, and also achieves better or comparable performance in terms of $K$-Fair. This suggests that minimizing ICRD can improve $K$-Fair. ii) ICCFL-wF exhibits a slight decrease in violation of model fairness. The reason is that the incorporation of interventional samples renders the training data more balanced across sensitive groups than the original dataset. iii) Baseline obtains the worst MMD score, which means its prediction distribution gap between different sensitive groups is the most evident. Because it optimizes classification loss without fairness objectives. ICCFL-KF regularizes the $K$-Fair metric on empirical risk, thus the prediction gap and MMD are small than Baseline. Nevertheless, it leaves a lot to be desired in terms of reducing predictive distributions disparity. In contrast, our ICCFL is much more effective in narrowing the gap between predictive distributions. iv) We preliminary observe that ICRD and MMD exhibit similar patterns of variation, with lower ICRD values aligning with smaller MMD values presented in ICCFL. Our hyper-parameter analysis experiments in Appendix further confirm this observation.

## 6.4 PERFORMANCE ON INCOMPLETE AND NOISY GRAPHS

To provide a clear understanding of the practical application of ICCFL, we conduct experiments on synthetic datasets with imperfect causal graphs, as a reliable complete partially DAG (CPDAG) can be easily derived using causal discovery methods. We first randomly generate two DAGs: one with 10 nodes and 20 edges, and another with 20 nodes and 40 edges. More details about synthetic datasets are given in Appendix. For each DAG, we randomly choose one node as the sensitive attribute, and denote the last node in the topological order as the outcome. Un-

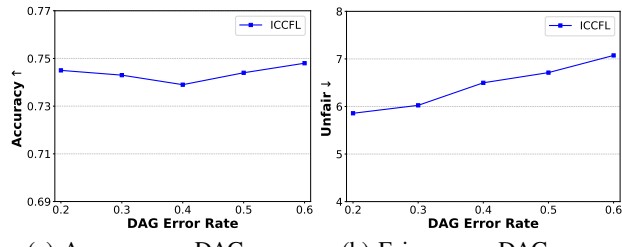

(a) Accuracy vs. DAG error    (b) Fairness vs. DAG error

Figure 3: Accuracy and fairness of ICCFL under different noisy DAG on Synthetic datasets (20 nodes and 40 edges).

der the premise of identifiability guarantee presented in Proposition 1, we generate two distinct partial DAGs, CPDAG1 and CPDAG2, by removing different numbers of edges. We also employ PC (Spirtes et al., 2001) to learn a CPDAG (denoted as CPDAG3). Figure 2 presents the performance of ICCFL with different CPDAGs, where ICCFL-true uses the ground-truth interventional data, ICCFL-1 refers to CPDAG1, ICCFL-2 leverages CPDAG2, and ICCFL-PC builds on CPDAG3 generated by PC. We observe that. (i) ICCFL effectively handles incomplete causal graphs and exhibits resilience to varying quality of CPDAGs. This is confirmed by the limited performance gaps between ICCFL-PC and others. (ii) As long as the critical directed edges, conforming to the identifiability criteria in Proposition 1, remain in CPDAGs, the causal interactions of other variables do not impact the efficacy of ICCFL, because the same causal identification formulas in Proposition 1 can still be used.

To evaluate the performance of ICCFL with respect to noisy DAGs, we also investigate the impact of $\beta$-level error rate (from 0.2 to 0.6) of DAGs on ICCFL, where such noisy DAGs are created by randomly removing or reversing directed edges in the true DAG (20 nodes and 40 edges). As can be seen in Figure 3, as the error rate of DAGs increases, our ICCFL shows a slight drop in fairness performance, demonstrating that ICCFL is robust to noise in DAGs.

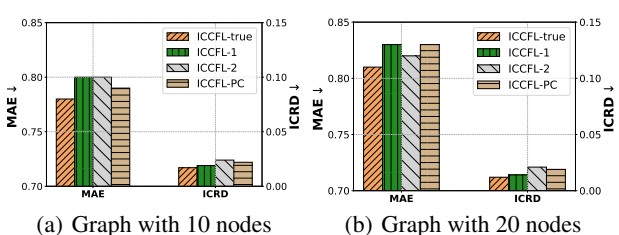

(a) Graph with 10 nodes    (b) Graph with 20 nodes

Figure 4: Accuracy and fairness trade-offs of ICCFL under unidentifiable cases.

We further testify ICCFL in cases where ICRD is unidentifiable on incomplete graphs. More details can be seen in Appendix. The results shown in Figures 2 and 4 confirm our ICCFL can effectively handle unidentifiable scenarios by compiling all possible CPDAGs that align with identifiability criteria of Proposition 1.

## 7 CONCLUSIONS

In this paper, we first uncovered the limitations of existing interventional fairness notions, revealing that these fairness notions often fall short in capturing the unfair causal effects of sensitive attributes on outcomes. We then introduced a novel intervention fairness notion (ICRD) to measure the post-intervention cumulative causal effects along the prediction probabilities for any intervention on the context $do(\mathbf{C} = \mathbf{c})$. Subsequently, we presented a causality-based fairness framework (ICCFL) to approximately assess and reduce ICRD values for achieving causal fairness. Theory analysis and experiments on real-world datasets confirm the effectiveness of our ICRD and ICCFL.

## 8 ETHICS STATEMENT

We promise that we have read the ICLR Code of Ethics, and this paper has not raised any questions regarding the Code of Ethics.

## 9 REPRODUCIBILITY STATEMENT

We promise that `ICCFL` is reproducibility. An anonymous link to the downloadable source code is provided in the appendix. We also provide the experimental setting, especially for detailed hyper-parameters settings, in the main text and Appendix. We will make our code publicly available once the paper is published.

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

## A   THE USE OF LARGE LANGUAGE MODELS (LLMs)

We declare that LLMs do not play a significant role in research ideation and/or writing.

## B   METHODOLOGY

### B.1   ICCFL ALGORITHEM

The overall procedure of ICCFL is presented in Algorithm 1. Lines 1-2 generate interventional samples based on causal model $\mathcal{M}$. Subsequently, at each epoch $t$, Line 5 computes the gradients of the model parameters for each sample with a mini-batch, and Line 6 updates the model parameters to reduce unfair cumulative unfair effects caused by the sensitive attribute.

---

**Algorithm 1** ICCFL: Intervention-based Cumulative Causal Fairness Learning

---

**Input**: The training data $\mathcal{D} = \{(s^i, \mathbf{x}^i, y^i) | 1 \leq i \leq n\}$, Causal Model $\mathcal{M}$, hyper-parameters $\lambda$ and $\tau$, learning rate $\eta$.
**Output**: Model parameters $\theta^*$
  1: Sample $u$ from the distribution $P(U|S = s, \mathbf{X} = \mathbf{x})$
  2: Generate interventional samples based on the sampled $u$ and causal model $\mathcal{M}$
  3: **for** epoch $t = 1, 2, \cdots, T$ **do**
  4:   **for** each mini-batch $\mathcal{B} \subseteq \mathcal{D}$ **do**
  5:     Compute $\nabla_\theta \mathcal{L} = \nabla_\theta (\frac{1}{|\mathcal{B}|} \sum_{i=1}^{|\mathcal{B}|} \ell(\tilde{y}, \tilde{y}^i) + \lambda \widehat{\text{ICRD}})$
  6:     $\theta_{t+1} \leftarrow \theta_t - \eta \nabla_\theta \mathcal{L}$
  7:   **end for**
  8: **end for**
  9: **return** model parameters $\theta^*$

---

### B.2   HANDLING MULTIPLE SENSITIVE ATTRIBUTES

In this subsection, we show how our ICCFL can easily extend to multiple sensitive attributes. When there are multiple sensitive attributes $S_1, \cdots, S_m$, two similar methods can be applied. First, we can consider $F(\tilde{y}|do(S_k = s_k), do(C = c))$ for each sensitive attribute $S_k$ as different constraints, i.e., we require that $\forall k, \int_0^1 F(\tilde{y}|do(S_k = s_k), do(C = c)) - F(\tilde{y}|do(S_k = s'_k), do(C = c)) \leq \mu$ where $\mu$ is the fairness tolerance parameter. On the other hand, we can consider the combination of all sensitive attributes, i.e., we consider $F(\tilde{y}|do(S_1 = s_1, \cdots, S_m = s_m), do(C = c))$ for all sensitive attributes.

### B.3   COMPUTATION COMPLEXITY

Compared to existing notions (e.g., $K$-Fair), the dominating computational cost of assessing ICRD is numerical integration with $M$ probing points, and the cost scales linearly with $M$. The main time complexity of ICCFL is $\mathcal{O}(Tn(lq^2 + (d + M)q)$, where $n$ is the sample size with $d$-dimensional variables, $l$ is the number of network layers, each of which has $q$ neurons. Besides, generating interventional samples scales linearly with the number of samples. Therefore, ICCFL is feasible for large datasets.

## C   EXPERIMENTS

### C.1   SYNTHETIC DATASET

In this section, we introduce the generation of synthetic datasets. Specifically, we first randomly generate two DAGs according to the Erdos-Renyi (ER) model: one with 10 nodes and 20 edges, and another with 20 nodes and 40 edges. For each DAG, we randomly choose one node as the sensitive attribute, and denote the last node in the topological order as the outcome. We assume that the data generation mechanism follows a linear additive noise model, with the structural function for

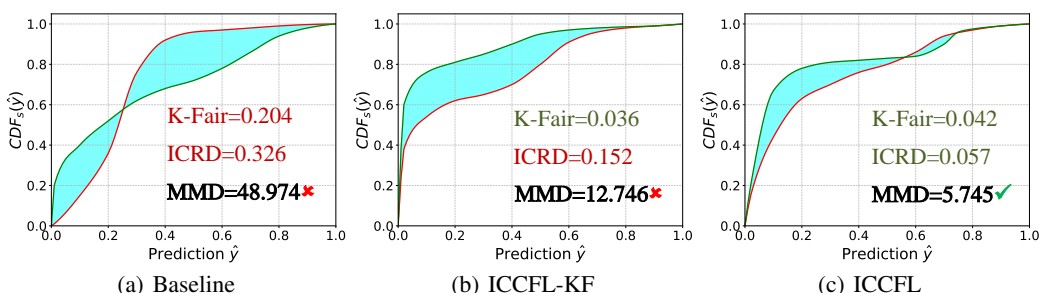

(a) Baseline        (b) ICCFL-KF        (c) ICCFL

Figure S1: The empirical cumulative distribution functions (CDFs) of the predictive probability over different sensitive groups (i.e., male and female) on the Adult dataset. (a) is the baseline method which uses all variables to train the model without considering the fairness; (b) is the variant of our `ICCFL`, named `ICCFL`-KF, which egularizes the $K$-Fair metric on empirical risk; and (c) is our `ICCFL`.

each node defined as $V_j = \sum_{V_i \in Pa(V_j)} \omega_{ji} V_i + \epsilon_j$, where $\omega_{ji}$ is randomly drawn from a Uniform distribution, and $\epsilon \sim N(0, 1)$. Then we generate 2000 samples where each node is assigned with two or three possible values.

Table S1: Method specific hyper-parameters: $lr$ is the learning rate of the corresponding model, $\lambda$ is the parameter of the fairness constraint (ALCF). For $\epsilon$-IFair, $\lambda$ is the fairness weight that controls the trade-off between accuracy and fairness. SeqSel and FairCFS methods do not have any specific hyper-parameters that require tuning.

| Method | Hyper-parameters |
|---|---|
| BL | $lr \in \{0.001, 0.002, 0.005, 0.01, 0.02, 0.05, 0.1, 0.2, 0.5\}$ |
| Unaware | $lr \in \{0.001, 0.002, 0.005, 0.01, 0.02, 0.05, 0.1, 0.2, 0.5\}$ |
| A3 | $lr \in \{0.001, 0.002, 0.005, 0.01, 0.02, 0.05, 0.1, 0.2, 0.5\}$ |
| ALCF | $\lambda \in \{0.0, 0.2, 0.4, 0.6, 0.8\}$ |
| $\epsilon$-IFair | $\lambda \in \{0.5, 1, 2, 5, 10, 20, 60\}$ |
| `ICCFL` | $\lambda \in \{0.05, 0.5, 1.0, 2, 5, 10, 20\}$, $\tau \in \{5, 10, 20, 30, 50\}$ |

## C.2 The Superiority of ICRD

Figure S1 shows the cumulative distribution functions of model predictions for sensitive groups on Adult dataset. As can be seen, Baseline obtains the worst prediction distribution gap, since it optimizes classification loss without fairness objectives. `ICCFL`-KF regularizes the $K$-Fair metric on empirical risk, thus the prediction gap is smaller than Baseline. However, `ICCFL`-KF leaves a lot to be desired in terms of reducing predictive distributions disparity. In contrast, our `ICCFL` is much more effective in narrowing the gap between predictive distributions.

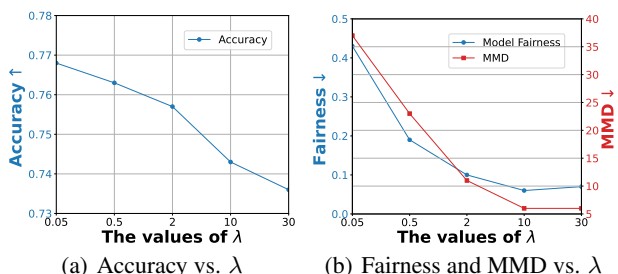

(a) Accuracy vs. $\lambda$     (b) Fairness and MMD vs. $\lambda$

Figure S2: Performance of `ICCFL` vs. hyper-parameter $\lambda$.

## C.3 Hyper-parameter Settings

Each compared method uses the same two hidden layers ReLU neural network with 64 hidden neurons as the base model. Therefore, they have the same number of model parameters. We use the

Table S2: The detailed hyper-parameters of compared methods on each dataset

|  |  | Adult | Dutch | Law School | ACSPublicCoverage |
|---|---|---|---|---|---|
| BL | Learning rate $lr$ | 0.002 | 0.001 | 0.001 | 0.01 |
| Unaware | Learning rate $lr$ | 0.002 | 0.001 | 0.001 | 0.01 |
| A3 | Learning rate $lr$ | 0.001 | 0.001 | 0.001 | 0.01 |
| ALCF | Learning rate $lr$ | 0.001 | 0.001 | 0.001 | 0.005 |
|  | Fairness constraint $\lambda$ | 0.4 | 0.2 | 0.4 | 0.6 |
| $\epsilon$-IFair | Learning rate $lr$ | 0.002 | 0.002 | 0.001 | 0.01 |
|  | Fairness weight $\lambda$ | 2 | 1 | 1 | 2 |
| ICCFL | learning rate $lr$ | 0.002 | 0.001 | 0.001 | 0.01 |
|  | Fairness constraint $\lambda$ | 1 | 1 | 1 | 1 |
|  | Temperature $\tau$ | 10 | 5 | 10 | 10 |

grid search strategy on the validation set to find the best hyper-parameters for all compared methods. We verify all methods with their hyper-parameters as listed in Table S1. In addition, the detailed hyper-parameters of compared methods on each dataset are summarized in Table S2.

## C.4 HYPER-PARAMETER ANALYSIS

**Impacts of $\lambda$.** In our proposed ICCFL, $\lambda$ is a crucial hyper-parameter that controls the trade-off between accuracy and fairness. As such, in this section, we conduct experiments on Adult dataset (similar patterns can be observed on other datasets) to analyze the impact of $\lambda$ by varying it within $\{0.05, 0.5, 2.0, 10, 30\}$ and report the results in Figure S2 and Figure S3. We can observe that. i) As expected, when $\lambda$ increases, ICCFL places greater emphasis on model fairness. As a result, ICCFL achieves a better causal fairness at the expense of lower accuracy. ii)

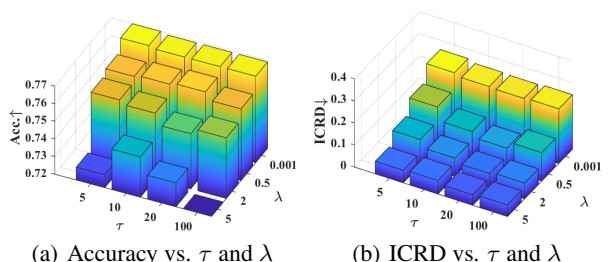

(a) Accuracy vs. $\tau$ and $\lambda$    (b) ICRD vs. $\tau$ and $\lambda$

Figure S3: Accuracy and fairness trade-offs as $\tau$ and $\lambda$ vary.

ICRD and MMD exhibit similar trends, with a decrease in ICRD aligning with a reduction in MMD. This positive correlation suggests that as the ICRD value downgrades, the model's predictions become increasingly fair for sensitive groups, consistent with the Property 3 outlined in Theorem 1. iii) In practice, setting $\lambda$ within the range $[1, 5]$ typically yields a favorable trade-off between fairness and accuracy.

**Impacts of $\tau$.** The hype-parameter $\tau$ in our proposed ICCFL is also crucial to approximate the true post-intervention cumulative causal effects. To verify the impact of this hype-parameter, we also conduct experiments on Adult dataset by varying $\tau$ within $\{3, 10, 20, 100\}$. The corresponding results are shown in Figure S3. We can observe that: i) As $\tau$ increases, the evaluation error of proposed ICRD decreases. This is in line with Theorem 2. ii) When $\tau$ is too large (e.g., $\tau$=100), the gradient may vanish, thereby restricting the model's learning capacity and hindering convergence during model updating. iii) The moderate values of $\tau$ (e.g., $\tau$=10 or $\tau$=20) are recommended to effectively balance the model performance and the gradient problem during optimization.

## C.5 ROBUSTNESS ON NOISY DAGS

To further evaluate the performance of our ICCFL with respect to noisy DAGs, we investigate the impact of $\beta$-level error rate of DAGs on ICCFL, where such noisy DAGs are created by randomly removing or reversing directed edges in the true DAG (20 nodes and 54 edges). The corresponding results are shown in Figure 3. As can be seen, as the error rate of DAGs increases, our ICCFL shows a slight drop in fairness performance, demonstrating that ICCFL is robust to noise in DAGs.

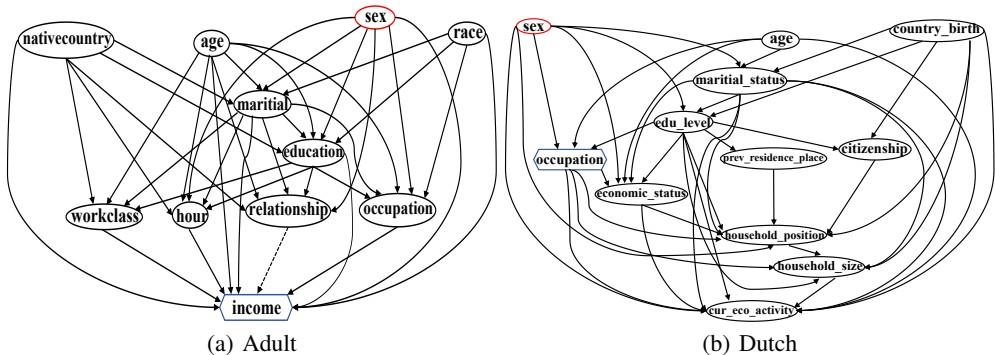

(a) Adult      (b) Dutch

Figure S4: The ground truth causal graphs of the Adult and Dutch datasets.

## C.6 HANDLING UNIDENTIFIABLE CASES

We further testify ICCFL when causal effects are unidentifiable on incomplete causal graphs, in which CPDAG fails to satisfy the identifiability criteria. We use the same synthetic datasets mentioned in Section 6.4 with the same settings, we create two unidentifiable CPDAGs, uCPDAG1 and uCPDAG2, and also apply PC to generate a CPDAG without additional knowledge. Figure 4 presents the results of fairness and predictions under different CPDAGs. From Figure 2 and 4, we can observe that ICCFL is still able to effectively balance fairness and accuracy in unidentifiable scenarios of causal effects. Furthermore, predictive models learned from all possible CPDAGs do not necessarily have a lower performance than those based on identifiable causal graphs.

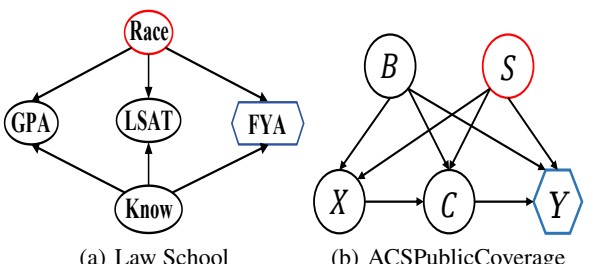

(a) Law School      (b) ACSPublicCoverage

Figure S5: (a) The ground truth causal graph for the Law School dataset; (b) The ground truth causal graph for the ACSPublicCoverage dataset where the node $S$ denotes the sensitive attribute, the node $B$ is introduced for base variables which are independent of $S$, the node $C$ is introduced for admissible contexts, the node $X$ are those variables which are affected by $S$ and the node $Y$ denotes the outcome.

## C.7 THE SOURCE CODE OF ICCFL

The code implementation of our ICCFL is available at `https://anonymous.4open.science/r/Intervention-based-Cumulative-Causality-Fairness-Learning-1D2D`.

## D THE CAUSAL GRAPHS OF REAL-WORLD DATASETS

We consider the causal graph introduced by Kusner et al. (2017) (level-2 causal model) as the ground truth for the Law School dataset and that introduced by Helwegen et al. (2020) (level-3 causal model) as the ground truth for the ACSPublicCoverage dataset. Figure S5 shows the ground-truth causal graphs of the Law School and the ACSPublicCoverage datasets. Figure S4 shows the ground-truth causal graphs of the Adult dataset and the Dutch dataset, which are constructed following (Zhang et al., 2018; Wu et al., 2019). Specifically, we employ the PC algorithm (Spirtes et al., 2000) and set the significance threshold 0.01 for conditional independence testing in graph construction.

# E   THE PROOF OF THEOREM 1

**❶ The proof of Property 1:**
If the model predictions satisfy causal fairness, the predictive probabilities under different interventions on the sensitive attribute should be the same. That is to say, given any different interventions on the sensitive attribute, the post-intervention distributions of the predictive probability conform to the identical distribution, i.e., $\forall \tilde{y} \in [0,1], F(\tilde{y}|do(S = s^+), do(\mathbf{C} = \mathbf{c})) = F(\tilde{y}|do(S = s^-), do(\mathbf{C} = \mathbf{c}))$. Then, according to Eq. (2) and Eq. (3), we can obtain $\mathrm{ICRD}(\tilde{y}) = 0$.

Conversely, according to the Definition 1 for $\mathrm{ICRD}(\tilde{y})$, the following holds:

$$
\begin{aligned}
& \mathrm{ICRD}(\tilde{y}) = 0 \\
& \Rightarrow F(\tilde{y}|do(S = s^+), do(\mathbf{C} = \mathbf{c})) \\
& \quad = F(\tilde{y}|do(S = s^-), do(\mathbf{C} = \mathbf{c})), \\
& \quad \forall \tilde{y} \in [0,1]
\end{aligned}
\tag{S1}
$$

Therefore, ICRD is a sufficient and necessary condition for the causal fairness:

$$
\begin{aligned}
& \mathrm{ICRD}(\tilde{y}) = 0 \\
& \Leftrightarrow F(\tilde{y}|do(S = s^+), do(\mathbf{C} = \mathbf{c})) \\
& \quad = F(\tilde{y}|do(S = s^-), do(\mathbf{C} = \mathbf{c})), \\
& \quad \forall \tilde{y} \in [0,1]
\end{aligned}
\tag{S2}
$$

**❷ The proof of Property 2:**
If the decision model is causal fairness, i.e., $\forall \tilde{y} \in [0,1], F(\tilde{y}|do(S = s^+), do(\mathbf{C} = \mathbf{c})) = F(\tilde{y}|do(S = s^-), do(\mathbf{C} = \mathbf{c}))$, then $\mathrm{ICRD}(\tilde{y}) = 0$. Besides, without loss of the generality, let $F_{s^+} = \arg\max F(\tilde{y}|do(S = s^+), do(\mathbf{C} = \mathbf{c})) = 1$ and $F_{s^-} = \arg\min F(\tilde{y}|do(S = s^-), do(\mathbf{C} = \mathbf{c})) = 0$, then we can obtain $\mathrm{ICRD}(\tilde{y}) = 1$. Thus, we have $\mathrm{ICRD}(\tilde{y}) \in [0,1]$.

**❸ The proof of Property 3:**
It is easy to verify the continuity condition of ICRD, as the estimation of the cumulative distribution function is continuous with respect to the model predictions, and our proposed fairness metric $\widehat{\mathrm{ICRD}}$ is also continuous with respect to the estimations of the cumulative distribution function.

# F   THE PROOF OF THEOREM 2

For any $\tilde{y} \in [0,1]$, we can obtain

$$
\lim_{\tau \to \infty} \sigma(\tilde{y} - \tilde{y}^i) = \frac{1}{1 + \exp(-\tau(\tilde{y} - \tilde{y}^i))} = \begin{cases} 1 & \text{if } \tilde{y}^i < \tilde{y}, \\ \frac{1}{2} & \text{if } \tilde{y}^i = \tilde{y}, \\ 0 & \text{if } \tilde{y}^i > \tilde{y}, \end{cases}
\tag{S3}
$$

Then under any intervention on the sensitive attribute and contexts $(do(S = s), do(\mathbf{C} = \mathbf{c}))$, we have

$$
\begin{aligned}
\lim_{\tau \to \infty} \sum_{i=1}^{n} \sigma_\tau(\tilde{y} - \tilde{y}_{S \leftarrow s}) & = \sum_{i=1}^{n} \lim_{\tau \to \infty} \sigma_\tau(\tilde{y} - \tilde{y}_{S \leftarrow s}) \\
& = \sum_{i=1}^{n} \mathbb{I}(\tilde{y}_{S \leftarrow s} \leq \tilde{y})
\end{aligned}
\tag{S4}
$$

According to Eq. (S4), we can obtain

$$
\begin{aligned}
\lim_{\tau \to \infty} \widehat{\mathrm{ICRD}}(\tilde{y}) & = |\frac{1}{n_+} \sum_{i=1}^{n_+} \mathbb{I}(\tilde{y}_-^i \leq \tilde{y}) - \frac{1}{n_-} \sum_{i=1}^{n_-} \mathbb{I}(\tilde{y}_-^i \leq \tilde{y})| \\
& = \mathrm{ICRD}(\tilde{y})
\end{aligned}
\tag{S5}
$$

## G  THE PROOF OF PROPOSITION 1

We begin with the concept of partial causal ordering (PCO) (Perkovic, 2020), which forms the foundation of our identifiability criteria.

**Definition 2** (Partial Causal Ordering (PCO)). *Suppose $\mathcal{G} = (\mathbf{V}, \mathbf{E})$ is a CPDAG, where $\mathbf{V} = \{\mathbf{S}, \mathbf{X}, Y\}$. An ordering, $<$, of mutually disjoint sets of nodes $(\mathbf{B}_1, \cdots, \mathbf{B}_t) \subseteq \mathbf{V}$ in $\mathcal{G}$, where $t \geq 1$ and $\cup_{i=1}^{t} \mathbf{B}_i = \mathbf{V}$, is defined as a partial causal ordering if it satisfies the condition that there is at least an edge directed from $B_i \in \mathbf{B}_i$ to $B_j \in \mathbf{B}_j$ in $\mathcal{G}$ if $\mathbf{B}_i < \mathbf{B}_j$. For instance, consider a CPDAG $\mathcal{G}$ with $X_1 - X_2 \to X_3$, the $PCO(\{X_1, X_2, X_3\}, \mathcal{G}) = (\{X_1, X_2\} < \{X_3\})$.*

Next, we introduce the following proposition and lemma presented by Perkovic (2020), which is helpful in the proof of our Proposition 1:

**Proposition 2** (Unidentifiable conditions for causal effects (Perkovic, 2020).). *Suppose $\mathbf{X}$ and $\mathbf{Y}$ are the disjoint node sets in a MPDAG $\mathcal{G} = (\mathbf{V}, \mathbf{E})$. If there is a proper causal path from $\mathbf{X}$ to $\mathbf{Y}$ that starts with an undirected edge in $\mathcal{G}$, then the causal effect of $\mathbf{X}$ on $\mathbf{Y}$ is not identifiable in $\mathcal{G}$.*

Proposition 2 illustrates the necessary condition for the unidentifiability of the causal effects within MPDAGs.

**Lemma 1** (Rules of $do$-operator in MPDAG (Perkovic, 2020).). *Suppose that $\mathbf{S}$ and $\mathbf{Y}$ are disjoint node sets in $\mathbf{V}$ in a MPDAG $\mathcal{G} = (\mathbf{V}, \mathbf{E})$ and that there is no proper causal path from $\mathbf{S}$ to $\mathbf{Y}$ that with an undirected edge in $\mathcal{G}$. Let $[\mathcal{G}]$ be the set of all DAGs represented by $\mathcal{G}$, and $PCO(An(\mathbf{Y}, \mathcal{G}_{\mathbf{V} \backslash \mathbf{S}}), \mathcal{G}) = (\mathbf{B}_i, \cdots, \mathbf{B}_t)$, where $An(\mathbf{Y}, \mathcal{G})$ represents the set of ancestors of $\mathbf{Y}$ in $\mathcal{G}$ and $t \geq 1$. Then, the following rules hold:*
*(i) For $i \in \{2, \cdots, t\}$, let $\mathbf{P}_i = (\cup_{j=1}^{i-1} \mathbf{B}_i) \cap Pa(\mathbf{B}_i, \mathcal{G})$. Then for each DAG in $[\mathcal{G}]$ and each post-intervention density $f$ consistent with DAG, we have:*

$$f(\mathbf{b}_i | \mathbf{b}_{i-1}, \cdots, \mathbf{b}_1, do(\mathbf{s})) = f(\mathbf{b}_i | \mathbf{p}_i, do(\mathbf{s})) \tag{S6}$$

*(ii) For $i \in \{2, \cdots, t\}$, let $\mathbf{P}_i = (\cup_{j=1}^{i-1} \mathbf{B}_i) \cap Pa(\mathbf{B}_i, \mathcal{G})$. For $i \in \{1, \cdots, t\}$, let $\mathbf{S}_{\mathbf{p}_i} = \mathbf{S} \cap Pa(\mathbf{B}_i, \mathcal{G})$. Then, for each DAG in $[\mathcal{G}]$ and each post-intervention density $f$ consistent with DAG, we have:*

$$f(\mathbf{b}_i | \mathbf{p}_i, do(\mathbf{s})) = f(\mathbf{b}_i | \mathbf{p}_i, do(\mathbf{s}_{\mathbf{p}_i})) \tag{S7}$$

*and*

$$f(\mathbf{b}_i | \mathbf{p}_i, do(\mathbf{s}_{\mathbf{p}_i})) = f(\mathbf{b}_i | pa(\mathbf{b}_i, \mathcal{G}) \delta_{\mathbf{S}=\mathbf{s}}) \tag{S8}$$

Next, we provide the formal proof of Proposition 1 as follows:

① *Proof sufficiency condition.* According to the theorem established by Zuo et al. (2024) (Theorem 4.1 in their literature), an augmented graph $\mathcal{G}^* = \mathcal{G} \cup \{\hat{Y}\}$, combining a MPDAG $\mathcal{G}$ with model prediction $\hat{Y}$, is a MPDAG, where the new node $\hat{Y}$ is a child of all variables that affect the output of the predictive model. As a result, $PCO(An(\mathbf{V}', \mathcal{G}^*), \mathcal{G}^*) = PCO(\mathbf{V}', \mathcal{G}^*)$, where $An(\mathbf{V}', \mathcal{G}^*)$ represents the set of ancestors of $\mathbf{V}'$ in $\mathcal{G}^*$, and $\mathbf{V}' = \mathbf{V} \backslash \{\mathbf{S}, \mathbf{C}\}$. Since there is no any possibly causal relationship between $O \in \mathbf{S} \cup \mathbf{C}$ and $V \in \mathbf{V} \backslash \{\mathbf{S} \cup \mathbf{C}\}$ such that $O - V$ is in $\mathcal{G}^*$, there is no any pair of nodes $O \in \mathbf{S} \cup \mathbf{C}$ and $V \in \mathbf{V} \backslash \{\mathbf{S} \cup \mathbf{C}\}$ such that $O \in \mathbf{B}_i$ and $V \in \mathbf{B}_i$ simultaneously, for all $i = 1, \cdots, t$. Thus, $PCO(\mathbf{V}', \mathcal{G}^*) = PCO(\mathbf{V}, \mathcal{G}^*) \backslash \hat{\mathbf{B}}$ where $\hat{\mathbf{B}}$ are those containing variables $O \in \mathbf{S} \cup \mathbf{C}$.

As such, without losing the generality, Let $(\mathbf{B}_1, \cdots, \mathbf{B}_t) = PCO(An(\mathbf{V}', \mathcal{G}^*), \mathcal{G}^*)$. For $i \in \{2, ..., t\}$, let $\mathbf{P}_i = (\cup_{j=1}^{i-1} \mathbf{B}_i) \cap Pa(\mathbf{B}_i, \mathcal{G})$. For $i \in \{1, \cdots, t\}$, let $\mathbf{S}_{\mathbf{p}_i} = \mathbf{S} \cap Pa(\mathbf{B}_i, \mathcal{G})$, and $\mathbf{C}_{\mathbf{p}_i} = \mathbf{C} \cap Pa(\mathbf{B}_i, \mathcal{G})$. Then, for any density function $f(\mathbf{x})$ consistent with $\mathcal{G}$ and the conditional

density $f(\hat{y}|\mathbf{x})$ consistent with $\mathcal{G}^*$, we have:

$$
\int_0^1 F(\tilde{y}|do(\mathbf{S}=\mathbf{s}), do(\mathbf{C}=\mathbf{c}))\mathrm{d}\tilde{y}
$$

$$
= \int_0^1 \int_0^{\tilde{y}} \int f(\mathbf{b}, \hat{y}|do(\mathbf{s}), do(\mathbf{c}))\mathrm{d}\mathbf{b}\mathrm{d}\hat{y}\mathrm{d}\tilde{y}
$$

$$
= \int_0^1 \int_0^{\tilde{y}} \int f(\hat{y}|\mathbf{b}, do(\mathbf{s}), do(\mathbf{c}))f(\mathbf{b}_1|do(\mathbf{s}), do(\mathbf{c}))\prod_{i=2}^t f(\mathbf{b}_i|\mathbf{b}_{i-1}, ..., \mathbf{b}_1, do(\mathbf{s}), do(\mathbf{c}))\mathrm{d}\mathbf{b}\mathrm{d}\hat{y}\mathrm{d}\tilde{y}
$$

$$
= \int_0^1 \int_0^{\tilde{y}} \int f(\hat{y}|\mathbf{v}', \mathbf{s}, \mathbf{c})f(\mathbf{b}_1|do(\mathbf{s}), do(\mathbf{c}))\prod_{i=2}^t f(\mathbf{b}_i|\mathbf{p}_i, do(\mathbf{s}), do(\mathbf{c}))\mathrm{d}\mathbf{b}\mathrm{d}\hat{y}\mathrm{d}\tilde{y}
$$

$$
= \int_0^1 \int_0^{\tilde{y}} \int f(\hat{y}|\mathbf{v}', \mathbf{s}, \mathbf{c})f(\mathbf{b}_1|do(\mathbf{s}_{\mathbf{p}_i}), do(\mathbf{c}_{\mathbf{p}_i}))\prod_{i=2}^t f(\mathbf{b}_i|\mathbf{p}_i, do(\mathbf{s}_{\mathbf{p}_i}), do(\mathbf{c}_{\mathbf{p}_i}))\mathrm{d}\mathbf{b}\mathrm{d}\hat{y}\mathrm{d}\tilde{y}
$$

$$
= \int_0^1 \int_0^{\tilde{y}} \int f(\hat{y}|\mathbf{v}', \mathbf{s}, \mathbf{c}) \prod_{\mathbf{b}_i \subseteq \mathbf{v}'} f(\mathbf{b}_i|pa(\mathbf{b}_i, \mathcal{G})\delta_{\mathbf{S}=\mathbf{s}, \mathbf{C}=\mathbf{c}})\mathrm{d}\mathbf{v}'\mathrm{d}\hat{y}\mathrm{d}\tilde{y}
$$

$$
\text{(S9)}
$$

Thus, ICRD is identifiable.

② *Proof necessity condition.* According to Proposition 2, the necessary condition for causal quantities $f(\mathbf{y}|do(\mathbf{S}=\mathbf{s}))$ to be identifiable is satisfied, if there is no any possible causal relations between $O \in \mathbf{S}$ and $Y \in \mathbf{Y}$ such that $O - Y$ is in MPDAG $\mathcal{G}$. Since ICRD is identifiable, i.e., causal quantities $\int_0^1 F(\tilde{y}|do(\mathbf{S}=\mathbf{s}), do(\mathbf{C}=\mathbf{c}))\mathrm{d}\tilde{y}$ is identifiable, there is no any causal path between $O \in \mathbf{S}\cup\mathbf{C}$ and $V \in \{\hat{Y}\} \cup \mathbf{V}\backslash\{\mathbf{S}\cup\mathbf{C}\}$ such that $O - V$ is in $\mathcal{G}^*$.

By combining ① and ②, we complete the proof of Proposition 1.

