# OpenReview forum: "Intervention-based Cumulative Causal Fairness Learning"
_ICLR.cc/2026/Conference — Submitted to ICLR 2026_

### Official Review · Reviewer_PBxk · 2025-10-28

**Soundness:** 2
**Presentation:** 1
**Contribution:** 1
**Rating:** 2
**Confidence:** 5

**Summary:**

The authors propose an approach for assessing and achieving causal fairness, based on a new metric labeled post-Intervention Cumulative Ratio Disparity. Key ideas include:

- Introducing a new metric (ICRD), which instead of comparing only expectations, looks at the distributional distance in terms of the L1 distance of cumulative functions,
- Introducing a computationally tractable relaxation of ICRD using sigmoid functions, which allows differentiable optimization,
- Introducing a result for partial identification of the causal fairness metric under partial causal knowledge (MPDAG).

**Strengths:**

(S1) The paper deals with an important and timely topic of causal fairness,

(S2) Using a distributional distance (rather than comparing only expectations) is an interesting direction to explore.

**Weaknesses:**

(W1)  Theoretical contributions are unclear: in current form, the paper does not convey important theoretical results.

Theorem 1:
For part (1) is the notion of causally independent even defined in the text?
For (2), isn’t the range of ICRD [0, 1] an obvious immediate consequence?
For (3), ICRD is stated to be continuous, _but without even specifying with respect to which variable_ continuity holds? One has to check the appendix for understanding the statement;
Furthermore, the ICRD notation seems abused; in Eq. (4), ICRD is an integral over $\tilde y$; in Eq. (5) and appendix, ICRD takes $\tilde y$ as an argument.

Theorem 2:
The validity of the sigmoid approximation is a well-known result, so this part does not really add much (it is questionable whether this should be framed as a theorem). Additionally, the inconsistency in notation is very alarming: $n_+, n_-$ used in Eq. (S5), $n_1, n^-$ in Eq. (6), $n_1, n_2$ in Eq. (7).

Proposition 1:
The proposition statement is almost identical to the cited previous work of Zuo et. al. and Perkovic et. al. Even the proof in the appendix follows this work almost verbatim.

(W2) Identification concerns not discussed: the result of Proposition 1 establishes identification under a specific assumption (no edge O — V). There is almost no discussion on what happens if this assumption is violated? How common is it to have this assumption satisfied, for any choice of admissible context C? This seems to be a very important point that is not emphasized.

(W3) Role of admissible context not discussed: ICRD definition uses a fixed value C=c; clearly, enforcing this constraint needs to happen for each value of C=c (which could be very many). This important practical challenge is not properly discussed.

(W4) Based on the above, the overall level of rigor in mathematical notation needs to be substantially higher.

(W5) The writing of the paper is poor. There are a number of typos and errors (e.g., adjectives/nouns for causal/causality and fair/fairness are repeatedly swapped; inconsistent notation, etc.). Also, some of the writing is imprecise: e.g.,

Line 39, “cannot distinguish between discriminatory and spurious correlations” -> this statement is conceptually confused and not grounded in existing literature. The distinction between “discriminatory” and “spurious” correlations is not a recognized or meaningful contrast in causal or fairness analysis. “Discriminatory” refers to normative or causal notions of unfair influence, whereas “spurious” refers to statistical confounding or non-causal association

Line 46, “counterfactual notions require full knowledge of causal model” -> this statement is false, check for instance
Shpitser, Ilya, and Judea Pearl. "What counterfactuals can be tested." arXiv preprint arXiv:1206.5294 (2012).

Eq. (2) -> path-specific fairness notation not even introduced? (e.g., how to define/interpret $\hat y_{s|\pi, s’|\bar \pi}$).

**Questions:**

See weaknesses.

---

> ### Author Response · Authors · 2025-11-21
>
> Thanks for reviewing our work and the constructive comments! We hope that our response have adequately address your concerns, and misunderstandings and clarify the contributions of our work.
>
> > **W1.1: Theorem 1: For part (1) is the notion of causally independent even defined in the text?**
>
> **Response:** causal independence is used to denote that **a model’s prediction is dependent to hypothetical interventions on the sensitive attribute.** Namely, model output should remain unchanged when evaluated on an observed sample and on its corresponding interventional versions.
>
> > **W1.2: Theorem 1, isn’t the range of ICRD [0, 1] an obvious immediate consequence?**
>
> **Response:** Although the boundedness of ICRD within [0,1] may appear evident from its normalized construction, **we present the formal argument for the sake of conceptual completeness.** More importantly, **there are semantics associated with the two endpoints.** Specifically, ICRD=0 corresponds exactly to the absence of unfair causal effect, and ICRD=1 reflects the maximal prediction disparity. Thus, even though the interval [0,1] is straightforward, **the endpoints align with the two conceptual extremes of causal fairness, which requires justification beyond the superficial numerical bound.**
>
> > **W1.3: Theorem 2: The validity of the sigmoid approximation is a well-known result, so this part does not really add much (it is questionable whether this should be framed as a theorem).**
>
> **Response: Theorem 2 establishes the extension specific to our setting.** Theorem 2 proves that our proposed estimation $\widehat{ICRD}$ tends to **converge to the true $ICRD$** as $\tau$ increases. It makes the proposed ICCFL **self-contained** and demonstrates that **ICCFL calculates the value of $ ICRD$ exactly**, enabling provable recovery of a necessary-and-sufficient causal fairness metric.
>
> > **W1.4: Proposition 1: The proposition statement is almost identical to the cited previous work of Zuo et. al. and Perkovic et. al. Even the proof in the appendix follows this work almost verbatim**
>
> **Response:** While Proposition 1 builds on the general line of identifiability results for MPDAGs established by Perković et al. and later adapted by Zuo et al., **its purpose and scope differ from the results in these works.**
>
> Zuo et al., 2024 explored the connection between the graph properties and the data distribution across MPDAG and its augmented graph with respect to $\hat{Y}$, and showed that the augmented graph qualifies an MPDAG. Their analysis enables the **identification of *Local* causal responses** on MPDAGs, i.e., the effect of an intervention on the outcome at a specific evaluation point.
>
> However, we are unable to directly adopt Zuo et. al. causal identification findings, because ICRD concerns a different quantity: it measures the **cumulative causal influence of the sensitive attribute across the entire predictive domain.** This is a distribution-level causal construct, not a pointwise causal effect. As such, the identifiability of local responses in Zuo et al. does not suffice to establish identifiability of such cumulative interventional disparities. To bridge this gap, **Proposition 1** extends the identifiability results of Zuo et al. to this domain-level setting. The proposition **provides the necessary and sufficient conditions under which the cumulative causal influence underlying ICRD is identifiable from an MPDAG**, which is not addressed by prior work.
>
> > **W2: the result of Proposition 1 establishes identification under a specific assumption (no edge O — V). There is almost no discussion on what happens if this assumption is violated? How common is it to have this assumption satisfied, for any choice of admissible context C?**
>
> **Response:** We would like to clarify that we do discuss the case in which the assumption of no undirected edge $O - V$ is violated (**Lines 308–317**). When such an edge is present, we can compile all MPDAGs that align with each valid permutation of edge orientations between $O - V$. As a result, we can uniquely identify the cumulative causal quantity in each MPDAG, and replace the fairness constraint with the average of unfairness over these MPDAG.
>
> **The identifying assumption is typically easy to satisfy in practice.** In most applications the set $O \in S \cup C$ (where admissible contexts $C$ are chosen based on users) contains only **a small number of variables**. Thus, the graphical criterion in Proposition 1 holds for the majority of realistic specifications.
>
> > **W4: typos and presentation issues**
>
> **Response:** We thank the reviewer for the careful reading and for pointing out the typo and presentation issues. We will correct all typos issues and improve the clarity and consistency of the presentation in the revised paper.

---

> ### Author Response · Authors · 2025-11-21
>
> > **W3: The important practical challenge of admissible context is not properly discussed**
>
> **Response:** The admissible contexts C denote those variables through which the sensitive attribute is permitted to influence the outcome. To make ICRD practical, the user is therefore allowed to classify variables into admissible and inadmissible, and thus, **a specific intervention on the admissible context is defined as part of the fairness specification itself.**
>
> **Practically, we operate on the observed data distribution: for each sample with its observed c, we generate its interventional counterpart (i.e., do(C=c)).** The optimization thus enforces fairness **relative to the user-specified admissible context**, rather than over an exhaustive enumeration of all possible context values, many of which may be irrelevant.
>
> If one considers multiple values of C, the fairness term in Objective can be replaced with the average of ICRD over different interventions on C.
>
> > **W5.1: imprecise writting: Line 39, “cannot distinguish between discriminatory and spurious correlations” -> this statement is conceptually confused and not grounded in existing literature.**
>
> **Response:** We thank the reviewer for this clarification. Our intended meaning was not to contrast “discriminatory” with “spurious” correlations, but rather to state that **correlation-based approaches cannot distinguish causal from spurious correlations between the sensitive attribute and the outcome. A classical illustration is Simpson’s paradox, where the statistical conclusions drawn from the sub-populations differ from that from the whole population.** Makhlouf et al. (2021) provide a concrete and intuitive discussion of how such spurious associations can mislead correlation-based fairness notions.
>
> [1] Makhlouf et al. Survey on Causal-based Machine Learning Fairness Notions. *arXiv:2010.09553v3*, 2021.
>
> > **W5.2: “counterfactual notions require full knowledge of causal model” -> this statement is false, check for instance Shpitser, Ilya, and Judea Pearl. "What counterfactuals can be tested." arXiv preprint arXiv:1206.5294 (2012).**
>
> **Response: The exact computation of counterfactual quantities requires full knowledge of causal model**, e.g., a fully specified causal graph. **Shpitser & Pearl (2012) do *NOT* contradict this point.** Their contribution is to identify the **graphical conditions under which counterfactuals are identifiable from observational data**, and they provide the IDC* algorithm to compute such counterfactual quantities for the given causal graph. Crucially, the IDC* algorithm **relies on full knowledge of the causal graph** and succeeds only when the required graphical criteria are satisfied.
>
> > **Eq. (2) -> path-specific fairness notation not even introduced? (e.g., how to define/interpret $\hat{y}_{s|\pi, s’|\bar{\pi}}$)**
>
> **Response:** We would like to respectfully clarify that the notion $\hat{y}_ {s|\pi, s'|\bar{\pi}}$ **is indeed introduced** in the paper: its definition and interpretation is given in **Lines 162–163**. $\hat{y}_{s|\pi, s’|\bar{\pi}}$ is the post-intervention outcome where the effect of intervention $do(s)$ is transmitted along $\pi$ while the effect of $do(s’)$ is transmitted along other paths.

---

> > ### Comment · Reviewer_PBxk · 2025-11-22
> >
> > I thank the authors for their response, which I have looked at carefully. However, most of my major concerns remain. The paper and review response use words in place of formal mathematical statements, which causes confusion in certain places.
> >
> > Specifical concerns that remain:
> >
> > (W1)
> >
> > Theorem 1: you seem to argue about "causally independent" with a mathematical statement. From your current framing, various different definitions could be written out: is it counterfactual fairness? zero total effect? identical distributions under do interventions? The level of rigor needs be higher, as already mentioned.
> >
> > Furthermore, while I appreciate your goal of "conceptual completeness", this does not seem to address the fundamental point, which is that one cannot see what the non-trivial part of the theorem statement is.
> >
> > Theorem 2: I understand the result on the sigmoid is adapted to your use case; however, this is not a major theoretical result.
> >
> > Proposition 1: I opened Zuo et. al. 2024 once more. Their work discusses identifiability of the densities $f(v | do(s)), f(y | do(s))$. Therefore, I do not understand your argument about their result being "pointwise". The proposed metric in your work is a functional of these densities?
> >
> > (W2) The justification for possible O - V edges does not seem principled. First, the fact that "few variables are in C practically" does not seem to be a defensible statement. Further, once a non-identifiable case is obtained, the mentioned approach of taking an average over "all graphs in class" has not really been investigated, and the implications of this case are unclear.
> >
> > (W3) The part on C=c seems every more concerning now. Are you implying that users are interested in a single (or very few) values of C=c generally? The notion of replacing the criterion with an average over different values of C=c has not been explored in the paper, which seems to be a drawback that could be addressed in future work.
> >
> > (W5) Once again, using words in place of formal statements results in imprecise claims. When you say "full causal model", one may think of the structural causal model -- in which case the statement is untrue, since a full SCM is not needed for computing a counterfactual.
> >
> > Furthermore, your claim about "full knowledge of the causal graph" is again not true -- this depends on the counterfactual that is studied. For instance, the effect of treatment on the treated is a counterfactual quantity that can be identified using only partial knowledge of the causal graph.
> >
> > Finally, your definition of path-specific in Line 162-163 is not a mathematical definition, but an intuitive description. See
> >
> > Avin, Chen, Ilya Shpitser, and Judea Pearl. "Identifiability of path-specific effects." (2005).
> >
> > for definitions.
> >
> >  ---
> >
> > Given the above points, I maintain my evaluation.

---

> ### Author Response · Authors · 2025-11-28
>
> We sincerely thank the reviewer for the timely follow-up and for highlighting the need for sharper mathematical articulation of key notions such as “causally independent’’ and path-specific responses, which is valuable for strengthening the presentation of the paper. But there are several points that we would like to further clarify.
>
> > the non-trivial part of the theorem 1 statement is.
>
> Theorem 1 is to articulate the desirable properties of the proposed ICRD. **The non-trivial part of the theorem is that it derives the first complete characterization of when fairness evaluation metrics imply causal fairness.** Specifically, we prove that ICRD=0 is a necessary and sufficient condition for causal fairness. This is the **first** formal proof that a fairness metric fully characterizes causal fairness, bridging a critical theoretical gap in the literature.
>
> > Proposition 1: I do not understand your argument about their result being "pointwise". The proposed metric in your work is a functional of these densities?
>
> Zuo et al. show that the augmented graph qualifies an MPDAG and, based on this, derive conditions under which the interventional density of the outcome at a specific evaluation point $\hat{Y}=\hat{y} $ (under a specific decision threshold), $f(\hat{Y}=\hat{y}|do(s))$ with the augmented variable \hat{Y}, is identifiable. Hence, “pointwise” refers to the value of the interventional density at the specific outcome point.
>
> ICRD is not defined at the level of a single point in the outcome space. It is a distribution-level functional that aggregates interventional disparities over the entire predictive domain. Its identifiability requires that the entire interventional distribution be uniquely determined across the MPDAG, not merely identifiable at isolated evaluation points.
>
> Proposition 1 fills this gap: it establishes identifiability criteria of cumulative interventional effects from an MPDAG. This requires extending Zuo et al. beyond local responses to global disparities over the outcome space.
>
> > W2
>
> The set C is the admissible contexts classified by users, i.e., variables through which users deem influence from the sensitive attribute to be legitimate within the application. Consequently, **whether an edge $O-V$ arises depends entirely on the task-specific admissible–inadmissible partition.** More importantly, the condition “no edge $O-V$” is *Not* introduced as an assumption. It is the **necessary and sufficient graphical criterion where ICRD is identifiable from an MPDAG.**
>
> Regarding the reviewer’s concern about non-identifiable cases, we clarify that **Section 6.4 and Appendix C.6** had already investigated this scenario empirically, whose results are shown in Fig. 4. We show that ICCFL is still able to effectively balance fairness and accuracy in unidentifiable scenarios of causal effects. Interestingly, **predictive models learned from all possible MPDAGs do not necessarily have a lower performance than those based on identifiable causal graphs.** (compared results between Fig. 2 and 4)
>
> > W3
>
> Once the admissible set C is fixed, ICRD can evaluate the disparity under any designated context C=c, which serves as the baseline for assessing interventional shifts. This does **Not** presuppose that users are interested in only a small number of context values; rather, it reflects the fact that fairness evaluation typically focuses on a chosen normative baseline (e.g., employees in the same department) rather than a full sweep over the entire C.
>
> Averaging ICRD over different interventions on C is a conceivable extension, as it could be attributed to the fact that they are penalizing different measures on fairness.
>
> > When you say "full causal model", one may think of the structural causal model -- in which case the statement is untrue, since a full SCM is not needed for computing a counterfactual.
>
> We appreciate the reviewer’s clarification. But a counterfactual can be computed without specifying all structural equations *only when* the causal graph satisfies the appropriate identifiability criteria, e.g., exogenous variables being mutually independent or decomposable so that they can be marginalized out. Thus, statements of the form “a full SCM is not needed” typically rely on these assumptions.
>
> > the effect of treatment on the treated is a counterfactual quantity that can be identified using only partial knowledge of the causal graph.
>
> The effect of treatment on the treated (ETT) is a variant of counterfactual fairness without conditioned on any non-sensitive attributes. According to Pearl’s factorization formula, **ETT also requires causal graph to identify**, i.e., $P(Y_s=y|s’)=\sum_{\mathbf{v’}}P(y|\mathbf{v}, s) \prod_{v \in \mathbf{v}\backslash s,y} P(v|pa(v)\delta_{S=s'})$ where $\mathbf{v’}$ denotes the variable set that satisfies the Backdoor Criterion relative to (S, Y), and $\delta_{S=s}$ denotes assigning variables in S involved in the term ahead with the corresponding values in s.

---

### Official Review · Reviewer_Hwj9 · 2025-10-29

**Soundness:** 3
**Presentation:** 2
**Contribution:** 3
**Rating:** 4
**Confidence:** 4

**Summary:**

This paper identifies the limitations of existing interventional fairness notions (e.g. K-Fair, Path-Specific Fairness), which fail to capture unfair causal effects of sensitive attributes on outcomes since they only compare average causal effects. The authors propose a new intervention-based fairness notion, post-Intervention-based Cumulative Ratio Disparity (ICRD), which measures distributional causal disparities through post-intervention cumulative prediction probabilities under contest $do(C=c)$. They further establish key theoretical properties and introduce a differentiable approximation to enable optimization. Experiments on both synthetic and real datasets demonstrate that the proposed ICCFL framework achieves better fairness-accuracy trade-offs and remains robust under noisy and incomplete causal graphs.

**Strengths:**

1. Significance: The paper clearly identifies key issues in prior intervention-based fairness notions, namely that expectation-based criteria may lead to inconsistent or insufficient assessments of causal fairness. It addresses these problems by providing a principled solution.
2. Originality: To the best of my knowledge, this is the first work that explicitly tackles the limitations of mean-based interventional fairness and introduces a distribution-level measure (ICRD) that accounts for the entire predictive distribution.
3. Quality: The paper conducts extensive experiments, including comparative studies and sensitivity analyses. The results consistently demonstrate the superiority of the proposed fairness metric (ICRD) and the robustness of the ICCFL framework across various datsets and causal graph settings.

**Weaknesses:**

1. The paper’s readability can be improved. There are several minor typos and presentation issues that slightly hinder the reading experience. For example:
    1. In line 191~193, the equation $P(\hat{y}=1|do(S=1), do(D=’A’)) - P(\hat{y}=1|do(S=0), do(D=’A’))=0.4$ seems to contain a typo, since $P(\hat{y}=1|do(S=1), do(D=’A’))=0.4$ and $P(\hat{y}=1|do(S=0), do(D=’A’))=0.2$.
    2. In line 205~206, the hyperlink for “Figure S1(b)” is not included, though I can guess which figure you want to mention.
    3. In Definition 1 (ICRD), the phrase 'a model is considered as causality fairness' should probably be “a model is considered causally fair.”
    4. In Equation (6), the term $\frac{1}{n-}$ seems to be a typo and should likely be $\frac{1}{n_2}$.
    5. In Proposition 1, the “augmented G” should be introduced or defined earlier for clarity.
2. Some parts of the method and analysis build on existing work (e.g. identifiability analysis and certain modeling components heavily rely on Zuo et al., 2024). While this does not undermine the contribution, it would be beneficial for the authors to better clarify which components are novel and which are inherited from previous frameworks.

**Questions:**

1. The paper refers to ICCFL as a “novel fair learning method,” but the main difference from Zuo et al. 2024 appears to lie only in the fairness regularization term. Could the authors clarify in what sense ICCFL represents a new *learning framework* rather than an existing pipeline with a new regularizer? Moreover, if the novelty mainly lies in the differentiable estimation of ICRD, why is the variant using K-Fair (i.e. ICCFL-KF) still called “ICCFL,” even though it does not employ the ICRD estimator?
2. The results show that both ICCFL-wF and ICCFL-KF outperform $\epsilon$-IFair, even though $\epsilon$-IFair also adopts a well-designed fairness regularizer within a similar training pipeline. Could the authors provide further analysis or intuition behind this phenomenon? How does this result support or contrast with the claimed advantages of ICRD and the ICCFL design?

---

> ### Author Response · Authors · 2025-11-21
>
> We greatly appreciate for your careful proofreading of our paper. Below we would like to address your questions and concerns point by point.
>
> > **W2: It would be beneficial for the authors to better clarify which components are novel and which are inherited from previous frameworks.**
>
> **Response:** We clarify below which parts of our framework are novel and which components are inherited from prior work.
>
> 1) Novel components
>
> Our main contribution lies in introducing ICRD, a novel intervention-based fairness notion that quantifies the integrated disparity over the entire predictive domain, thereby capturing distributional shifts between factual samples and their corresponding interventional ones. Most importantly, we are the *first* to provide theoretical analysis where **$ICRD=0$ is a necessary and sufficient condition for causal fairness.**
>
> In addition, we propose the **ICCFL optimization method**, which introduces temperature-scaled sigmoid function to enable differentiable optimization of interventional constraints.
>
> 2) Inherited yet substantially extended components.
>
> As discussed in the main text, our identifiability analysis indeed draws inspiration from Zuo et al., 2024. They explored the connection between the graph properties and the data distribution across MPDAG and its augmented graph with respect to $\hat{Y}$, which enables the formal identification of the causal effect on the MPDAG. While we inherit this foundational perspective, our focus is different:
>
> We extend the identifiability results to the **identifiability of cumulative causal interventional effects.** This extension requires new derivations because ICRD evaluates the cumulative causal influence of the sensitive attribute across the entire predictive domain, rather than the identification of a local causal response examined by Zuo et al., 2024.
>
> 3) Components inherited from previous work
>
> The generation of interventional data follows the procedure used in Zuo et al., 2024, which is a common adopted way for enforcing causal fairness.
>
> > **Q1: Could the authors clarify in what sense ICCFL represents a new learning framework rather than an existing pipeline with a new regularizer? Moreover, if the novelty mainly lies in the differentiable estimation of ICRD, why is the variant using K-Fair (i.e. ICCFL-KF) still called “ICCFL,” even though it does not employ the ICRD estimator?**
>
> **Response:** Thanks for this insightful comment! To clarify, the novelty of ICCFL does NOT lie solely in a new regularizer, but rather in the learning framework it induces. **ICCFL establishes a unified intervention-based fairness constraint learning paradigm through a differentiable approximation.** Specifically, ICCFL introduces a temperature-scaled sigmoid approximation to transform an interventional fairness constraint into a **differentiable optimization objective**, enabling gradient-based training for fairness notions. Within this paradigm, the differentiable formulation of ICRD serves as one instantiation of the framework, but the framework itself is not defined by this particular estimator.
>
> This brings us to the second question. **ICCFL-KF is still termed “ICCFL” because it adopts the same interventional-constraint learning framework**, differing only in the choice of the fairness quantity being regularized. K-Fair is also not differentiable with respect to the model parameters, and thus, it still requires a differentiable surrogate for optimization and fits into the same ICCFL pipeline without altering the underlying learning mechanism.

---

> ### Author Response · Authors · 2025-11-21
>
> > **Q2: The results show that both ICCFL-wF and ICCFL-KF outperform -IFair. Could the authors provide further analysis or intuition behind this phenomenon? How does this result support or contrast with the claimed advantages of ICRD and the ICCFL design?**
>
> **Response:** Thank you for this interesting and valuable comment.
>
> **ICCFL-wF** trains the predictive model without any fairness regularization. As a result, it naturally achieves **higher predictive accuracy** than IFair, but **at the cost of substantial fairness violations.**
>
> The performance gap between **ICCFL-KF** and IFair arises from **two fundamental differences** in how fairness is enforced:
>
> 1) **Interventional data generation.** Unlike IFair, ICCFL-KF generates the interventional versions for each sample, thereby constructing a **more balanced dataset** for evaluating interventional disparities.
>
> 2) **How discrepancy is optimized.** IFair employ Maximum Mean Discrepancy (MMD) to measure the discrepancy between $P(Y=1|do(S=s))$ and $P(Y=1|do(S=s’))$. In contrast, ICCFL-KF applies a **temperature-scaled sigmoid approximation** to obtain a differentiable surrogate of K-Fair, turning the interventional constraint into a trainable loss function. This enables a **smooth optimization** of the interventional fairness criterion. The superior performance of ICCFL-KF over -IFair shows that for many practical datasets, fairness constraints with a temperature-scaled sigmoid approximation is a **sweeter spot on the accuracy–fairness trade-off** than enforcing full matching between $P(Y=1|do(S=s))$ and $P(Y=1|do(S=s’))$.
>
> > **W1: There are several minor typos and presentation issues**
>
> **Response:** We thank the reviewer for the careful reading and for pointing out the typo and presentation issues. We will correct all typos issues and improve the clarity and consistency of the presentation in the revised paper.

---

### Official Review · Reviewer_8T6g · 2025-10-30

**Soundness:** 2
**Presentation:** 2
**Contribution:** 2
**Rating:** 4
**Confidence:** 4

**Summary:**

This paper identifies the limitations of existing interventional fairness notions and introduces a new causal fairness metric, Intervention-based Cumulative Rate Disparity (ICRD). ICRD measures the cumulative post-intervention causal effects along the prediction probabilities for any intervention on the context. In addition to formalizing this metric, the authors propose an algorithm to optimize for ICRD. Experimental results demonstrate the effectiveness of the proposed approach.

**Strengths:**

- The paper makes a meaningful attempt to analyze and reveal the weaknesses of existing interventional fairness notions and to propose a new one.

- The experimental evaluation is comprehensive, including practical settings with imperfect causal graphs.

**Weaknesses:**

1. While ICRD provides an interesting perspective, the contribution appears modest. The learning method proposed to achieve it is not particularly novel, and the paper would benefit from a clearer discussion of the situations where ICRD might be less suitable than alternative fairness notions.

2. In Theorem 1, the authors mention that ICRD is a continuous function. However, the benefit or implication of this property for ICRD is not discussed.

3. The identifiability analysis in Proposition 1 focuses primarily on MPDAG identification, which appears somewhat tangential to the main contribution of introducing ICRD.

4. Minor Issues / Typos:
- Please check the notation $s$ in Equation (3).
- Verify the denominator of $n_2$ in Equation (6).

**Questions:**

Please refer to the above weaknesses.

---

> ### Author Response · Authors · 2025-11-21
>
> Thanks for your valuable comments about our work! Below we would like to address your questions.
>
> > **W1: The contribution of ICRD, and the scenarios where ICRD might be less suitable than alternative fairness notions.**
>
> **Response: ICRD establishes foundational advances that address critical gaps in causal fairness literature.** Previous work defined causal fairness metrics as **heuristic criteria** that enforces necessary but not sufficient conditions for causal fairness. Thus, despite unfair effects in predictions, these metrics are satisfied.
>
> We derive **the first complete characterization of when fairness evaluation metrics imply causal fairness**, and propose a novel causal fairness notion ICRD, and delineate its **desirable properties** and **identification conditions**. Compared to other causal fairness metrics, e.g., counterfactual fairness, ICRD has minimal identifiability constraints. Most importantly, we demonstrate that **$ICRD=0$ is a necessary and sufficient condition for causal fairness.** This is the **first** formal proof that a fairness metric fully characterizes causal fairness, bridging a critical theoretical gap in the literature.
>
> As for the scenarios where ICRD may be less suitable. ICRD is not the preferred choice when the practitioner seeks to reason about *specific pathways* or enforce *path-specific fairness constraints*, in which case path-specific fairness may better match the policy objective. Besides, compared to other metrics, ICRD is **computationally more demanding**: the dominant cost arises from numerical integration over the predictive domain using $M$ probing points, but the runtime scales **linearly** with $M$.
>
> > **W2: the benefit or implication of the property of continuous for ICRD is not discussed.**
>
> **Response: The continuity of ICRD plays an important role in fairness evaluation.** Specifically, continuity ensures that small perturbations in the predictive distribution leads to proportionally small changes in the ICRD value. This property guarantees the **stability of the fairness judgments**, prevents pathological discontinuous jumps in the optimization landscape, and underpins **the validity of using a smooth surrogate (Theorem 2) to approximate ICRD during training.**
>
> > **W3: The identifiability analysis in Proposition 1 focuses primarily on MPDAG identification, which appears somewhat tangential to the main contribution of introducing ICRD**
>
> **Response:** We would like to respectfully clarify that **Proposition 1 does NOT focus on MPDAG identification.** Instead, it provides the **necessary and sufficient criterion under which ICRD is identifiable from an MPDAG**. Specifically, ICRD is identifiable iff MPDAG does not exhibit any possibly causal relationship between nodes $O\in S\cup C$ and nodes $V\in V∖(S\cup C)$, i.e., no undirected edge $O-V$.
>
> > **W4: typo issues**
>
> **Response:** We thank the reviewer for the careful reading and for pointing out the typo issues. We will correct all typos issues and improve the clarity and consistency of the presentation in the revised paper.

---

### Official Review · Reviewer_yVyi · 2025-10-30

**Soundness:** 3
**Presentation:** 3
**Contribution:** 2
**Rating:** 6
**Confidence:** 4

**Summary:**

The paper introduces a new causal-based ML fairness metric, which belongs to the category of intervention-based. This is motivated by two limitations of existing intervention-based metrics: Inconsistent assessment of bias and the insufficiency to conclude causal fairness. The new metric is called Intervention-based Cumulative Ratio Disparity (ICRD). The main idea behind this metric is that instead of relying on a specific threshold for the prediction (which might lead to equal averages between sensitive groups, despite discrimination), this metric considers the cumulative distribution function of the prediction. Along with the metric, the paper proposes a fair learning method (ICCFL) that mitigates bias as assessed by ICRD. The experimental analysis considers a significant number of existing causally fair learning methods. The comparison considers the Accuracy and fairness (measured using MMD, a distribution-level distance, to capture the divergence between distributions after intervention). In another set of experiments, ICRD is compared with other intervention-based metrics. The last set of experiments focused on the robustness of ICCFL in presence of noisy graphs.

**Strengths:**

The paper falls into the category of causal-based ML fairness approaches which is timely and very promising to address the problem of ML Bias.
The proposed ICRD fairness metric is well motivated and formally defined.
The experimental analysis involves a comparison with a representative set of existing causal-based approaches (which is not common in the literature, so a positive aspect of this work, but also raises some concerns as explained in the weaknesses).

**Weaknesses:**

1) My main concern with the ICRD fairness metric is its applicability in practice. As defined, it is non-differentiable, and I don't see how it can be reliably computed in practice. There is already major concerns about the practicality of Causal-based fairness metrics, with this proposed metric, it becomes more questionable.
2) Very few details were provided about how existing approaches have been implemented for comparison purposes (e.g. A3). I checked in the appendix and I didn't find further details about how this is exactly done. This is important to show in order to justify the claimed "superiority" of the approach.

Some minor issues:
Equation (1), pa is not defined (I assume it means parents, but not sure).
Equation (3) of K-Fair is flawed (the same terms on both sides).
Equation (6) there should be 2 instead of the dash in n^{-}
Propoosition 1: MPDAG is not defined unless it is a typo for CPDAG. For that proposition, I would suggest an intuitive illustration because it is too abstract to understand (at least in the appendix).
Other metrics can be included in the comparison, such as:
Pfohl, S. R., Duan, T., Ding, D. Y., & Shah, N. H. (2019, October). Counterfactual reasoning for fair clinical risk prediction. In Machine Learning for Healthcare Conference (pp. 325-358). PMLR.
Also, you can refer to this survey for other causal-based fairness metrics:
Makhlouf, K., Zhioua, S., & Palamidessi, C. (2024). When causality meets fairness: A survey. Journal of Logical and Algebraic Methods in Programming, 141, 101000.

**Questions:**

For Causal discovery, why still using PC while newer algorithms are much better (DiffAN / DeepDAG, etc.) ?

---

> ### Author Response · Authors · 2025-11-21
>
> We sincerely thank you for the helpful suggestions and positive assessment of our work’s contribution! Below, we hope to address your concern about our paper.
>
> > **W1: The concern about the applicability of ICRD fairness metric in practice.**
>
> **Response:ICRD is applicable in practice.** As detailed in Section 5.2, to make the metric trainable in practice, we introduce a temperature-scaled Sigmoid function that produces a differentiable approximation $\widehat{ICRD}$. Besides, our Theorem 2 provides a theoretical guarantee that $\widehat{ICRD}$ converges to $ICRD$ as the temperature increases, thereby ensuring the soundness of this approximation. Our hyperparameter study in Appendix C.4 further confirms this behavior empirically: larger temperatures reduce the evaluation error. However, excessively high temperatures can hinder training due to vanishing gradients. Therefore, a moderate range (such as $\tau=10$ to $\tau=20$) offers a stable trade-off for practical training.
>
> > **W2: The implemented details about A3 method.**
>
> **Response:** A3 assumes the causal model as additive noise model. Specifically, for each variable $V_i$, $P(V_i|pa(V_i))$ can be treated as $V_i = f_i(pa(V_i))+\epsilon_i$, where $pa(V_i)$ denotes the parents of $V_i$, and $\epsilon_i$ is the noise term. Following the topological ordering of the causal graph, we fit a regressor for each node using its parents as inputs and take the residuals as an estimate of $\epsilon_i$. Finally, $\hat{Y}$ is constructed using only the non-descendants of the sensitive attribute and their corresponding residuals.
>
> > **W3: Other metrics can be included in the comparison**
>
> **Response:** Thanks for the suggestion! We additionally include three intervention-based fairness notions as alternative variants of our method: **Total Causal Effect (TCE), No Proxy Discrimination (NPD), and Path-Specific Causal Fairness (PSCF)**. For path-specific fairness in particular, we follow Zhang et al. (2018): we use PSE-DD* to identify discriminatory path-specific effects and PSE-DR* to repair the dataset accordingly, after which a Demographic Parity constraint is applied during model training to enforce fairness. The results on Adult dataset are shown as follows.
>
> | |Acc.|ICRD|TCE|MMD|Acc.|ICRD|NPD|MMD|Acc.|ICRD|PSCF|MMD|
> |-|-|-|-|-|-|-|-|-|-|-|-|-|
> variant|0.745|0.097|0.044|9.462|0.743|0.158|0.032|13.094|0.739|0.088|0.024|8.716
> ICCFL|0.743|0.057|0.069|5.745|0.743|0.057|0.048|5.745|0.743|0.057|0.055|5.745
>
> As can be seen, these metrics can partially mitigate discriminatory behavior; however, they still **leave substantial discrepancies in the predictive distributions**. In contrast, ICCFL with ICRD **achieves smaller MMD values** between interventional predictive distributions, indicating that ICRD offers a more effective way to reduce causal unfairness at the distributional level.
>
> > **W4: typo issues:**
> Equation (1), pa is not defined (I assume it means parents, but not sure).
> Equation (3) of K-Fair is flawed (the same terms on both sides).
> Equation (6) there should be 2 instead of the dash in n^{-}
> MPDAG is not defined in Proposition 1
>
> **Response:**  We thank the reviewer for the careful reading and for pointing out the typo and presentation issues. Equation (1), pa is the parents of the variables. The right side of Eq. (3) should be $\mathbb{E}[\hat{y}|do(S=s’),do(\mathbf{C}=\mathbf{c})]$. $n^{-}$ should be $ n _2$ in Eq. (6).
>
> MPDAG refers to a **maximally oriented partially directed acyclic graph**, which is obtained by applying the Meek’s rules in Meek et al. 1995 to a CPDAG with a background knowledge constraint. Namely, a CPDAG provides the unique graphical representation of the Markov equivalence class of a DAG learned from causal discovery, and an MPDAG is derived from this CPDAG when additional domain knowledge is used to further orient edges. we will include the toy example causal graphs to show DAG, CPDAG and MPDAG.
>
> We will correct all typos and improve the clarity and consistency of the presentation in the revised paper.
>
> > **Q1: For Causal discovery, why still using PC while newer algorithms are much better (DiffAN / DeepDAG, etc.) ?**
>
> **Response:** For the purpose of ensuring direct comparability with prior studies, which rely on the PC-derived causal graph, we employ PC so that **our experiments are conducted under exactly the same structural assumptions.**
>
> [1] Zhang et al. Causal Modeling-Based Discrimination Discovery and Removal: Criteria, Bounds, and Algorithms. *IEEE TKDE*, 2018.
>
> [2] Meek et al. Causal inference and causal explanation with background knowledge. *Conference on Uncertainty in Artificial Intelligence*, 1995.

---

### Meta-Review · Area_Chair_giYD · 2025-12-30

**Summary:**

The paper introduces a new causal-based ML fairness metric, which belongs to the category of intervention-based.
The reviewers raised concerns regarding the clarity of the writing, the novelty of the proposed method, and the lack of mathematical rigor.

**Reviewer Concerns:**

The concerns of Reviewer PBxk seems to remain unaddressed, particularly regarding the novelty and mathematical rigor.

**Reviewer Scores:**

Responses to Reviewer yVyi and Reviewer 8T6g seem to have addressed their concerns, at least partially, so there might have been some increase in their scores.
Reviewer PBxk seems to have remained firm in their stance, so their score likely would not have changed.

---

### Decision · Program_Chairs · 2026-01-26

Reject